# 2-Hydroxyglutarate modulates histone methylation at specific loci and alters gene expression via Rph1 inhibition

Marios Gavriil[1], Marco Proietto[2] , Nicole Paczia[2], Aurelien Ginolhac[1] , Rashi Halder[2] , Elena Valceschini[1], Thomas Sauter[1] , Carole L Linster[2] , Lasse Sinkkonen[1]

**2-Hydroxyglutarate (2-HG) is an oncometabolite that accumulates in certain cancers. Gain-of-function mutations in isocitrate dehydrogenase lead to 2-HG accumulation at the expense of alpha-ketoglutarate. Elevated 2-HG levels inhibit histone and DNA demethylases, causing chromatin structure and gene regulation changes with tumorigenic consequences. We investigated the effects of elevated 2-HG levels in *Saccharomyces cerevisiae*, a yeast devoid of DNA methylation and heterochromatin-associated histone methylation. Our results demonstrate genetic background-dependent gene expression changes and altered H3K4 and H3K36 methylation at specific loci. Analysis of histone demethylase deletion strains indicated that 2-HG inhibits Rph1 sufficiently to induce extensive gene expression changes. Rph1 is the yeast homolog of human KDM4 demethylases and, among the yeast histone demethylases, was the most sensitive to the inhibitory effect of 2-HG in vitro. Interestingly, Rph1 deficiency favors gene repression and leads to further down-regulation of already silenced genes marked by low H3K4 and H3K36 trimethylation, but abundant in H3K36 dimethylation. Our results provide novel insights into the genome-wide effects of 2-HG and highlight Rph1 as its preferential demethylase target.**

## Introduction

There is a tight interplay between cellular metabolism and gene regulation at the chromatin level. Many of the chromatin-modifying enzymes use central metabolites as substrates or cofactors for catalysing the insertion ("writers") or removal ("erasers") of epigenetic marks, thereby sensing the metabolic state of the cell (Haws et al, 2020). Among the important "eraser" enzymes, TET family DNA demethylases and JmjC domain containing histone demethylases use alpha-ketoglutarate (a-KG), $Fe^{2+}$, and oxygen as substrates and cofactors for an oxidative reaction removing methyl groups from adenosine and cytosine in DNA and from lysine residues in various

histones, marking both active euchromatin and repressed heterochromatin regions (Pirozzi & Yan, 2021).

The activity of a-KG–dependent enzymes can be inhibited by metabolites such as 2-hydroxyglutarate (2-HG) that are structurally similar to a-KG (Xu et al, 2011). 2-HG is typically not abundant in higher eukaryotes, but can accumulate in some neurometabolic diseases and especially in certain cancers up to millimolar levels (Ye et al, 2018). The accumulation can be caused by loss-of-function mutations in enzymes metabolizing 2-HG, as is the case in most forms of 2-HG acidurias (Kranendijk et al, 2012), or by gain-of-function mutations in isocitrate dehydrogenases (IDHs), key enzymes of the TCA cycle (Chou et al, 2021). Heterozygous IDH mutations are present in various types of cancer such as leukemia and glioblastoma, where they are also used to molecularly classify the tumors (Molinaro et al, 2019). The recognition of 2-HG as an oncometabolite, whose accumulation may induce tumor-associated alterations in both DNA and histone methylation, triggered increased interest for this metabolite in the scientific community and intensive research efforts directed towards its metabolism and cellular effects (Pirozzi & Yan, 2021).

Because 2-HG can inhibit different demethylases, it has been a challenge to determine precisely which enzymes are responsible for the observed methylation changes and how they can lead to alterations in gene expression and to further phenotypic differences. For example, 2-HG–induced DNA methylation was shown to disrupt cohesin binding, thereby leading to loss of topological domain insulation and increased oncogene expression in IDH-mutant gliomas (Flavahan et al, 2016). Moreover, 2-HG accumulation can increase methylation at histone residues such as histone H3 lysine 4 (H3K4), lysine 36 (H3K36), and lysine 79 (H3K79), associated with active transcription, and repressive marks such as histone H3 lysine 9 (H3K9), lysine 27 (H3K27), and lysine 20 (H3K20) (Lu et al, 2012; Turcan et al, 2018). Among these, in particular, the increase in H3K9 methylation because of KDM4 demethylase inhibition has been linked to phenotypic changes such as decreased DNA damage repair (Inoue et al, 2016) and defective cellular differentiation (Lu et al, 2012; Juan-Manuel et al, 2019). At the same time, the contribution of altered chromatin methylation at active

---

[1]Department of Life Sciences and Medicine, University of Luxembourg, Belvaux, Luxembourg   [2]Luxembourg Centre for Systems Biomedicine, University of Luxembourg, Belvaux, Luxembourg

Correspondence: lasse.sinkkonen@uni.lu; carole.linster@uni.lu

marks such as H3K36 to the phenotypic changes has remained unclear, although increased H3K4 methylation has been associated with oncogene activation in glioma (Turcan et al, 2018).

The budding yeast *Saccharomyces cerevisiae* represents an interesting model organism for investigating the contributions of different 2-HG targets to the molecular changes induced upon 2-HG accumulation. It has recently been established that *S. cerevisiae* exclusively produces the D-enantiomer of 2-HG (Becker-Kettern et al, 2016) (in this article, 2-HG refers to D-2-HG except if stated otherwise), which is also the molecular species that accumulates in human tumors. *S. cerevisiae* is devoid of DNA methylation and repressive chromatin marks (Zhao & Garcia, 2015), but is well known to harbor three histone lysine residues targeted for methylation (H3K4, H3K36, and H3K79) and that can be demethylated by four JmjC domain-containing histone demethylases. Jhd2 is the sole histone demethylase known to act on methylated H3K4, whereas Jhd1, Gis1, and Rph1 are responsible for the demethylation of H3K36 (Separovich & Wilkins, 2021). In particular, Jhd1 can demethylate mono- and dimethylated H3K36 and Gis1 and its paralogue Rph1 can demethylate di- and trimethylated H3K36 (Tu et al, 2007). It has been shown that increased 2-HG levels in yeast can induce hypermethylation at two of the target residues, namely H3K4 and H3K36, and that inhibition of H3K36 demethylation can enhance gene silencing (Janke et al, 2017). However, the genetic loci affected by 2-HG accumulation and the impact on gene expression have not been investigated so far.

We have previously identified the *S. cerevisiae* Dld3 and Dld2 proteins as the enzymes responsible for metabolizing D-2-HG in the cytosol and mitochondria, respectively, of the yeast cell and generated deletion strains accumulating this metabolite at millimolar levels (Becker-Kettern et al, 2016). Here, we show that 2-HG can induce background-dependent transcriptome changes accompanied by locus-specific changes in H3K4 and H3K36 methylation that poorly correlate with the gene expression changes. Parallel analyses of demethylase deletion strains indicate that Rph1 inhibition is sufficient to account for the observed 2-HG-induced gene expression changes, in particular, through further silencing of genes already repressed by H3K36 methylation. Our results suggest that 2-HG more potently inhibits histone demethylases of the KDM4 family, leading to context-dependent outcomes on gene expression, and supports a role for H3K36 methylation in yeast heterochromatin silencing.

# Results

## 2-HG accumulation induces genetic background-dependent gene expression changes

To investigate the epigenomic and transcriptomic changes induced by 2-HG, we generated knockout strains accumulating 2-HG in two different genetic backgrounds, namely FY4 and BY4741 (Fig S1A). In our previous studies, Dld3 and Dld2 were identified as the enzymes responsible for the conversion of 2-HG to a-KG in yeast (Fig 1A) (Becker-Kettern et al, 2016). The BY4741 *dld3*Δ strain was used in Becker-Kettern et al (2016) and the accumulation of 2-HG was confirmed by liquid chromatography–mass spectrometry. Here, we

confirmed the accumulation of 2-HG in the FY4 *dld3*Δ strain (Fig S1B). In both genetic backgrounds, a 20-fold increase in the intracellular 2-HG concentration was observed in the *dld3*Δ knockout compared with the wild-type strain. The transcriptomic changes in both strains compared with their wild-type counterparts were investigated using RNA-seq (Fig 1B and C). For the RNA-seq experiments, five biological replicates were used for the FY4 strains and three biological replicates for the BY4741 strains. Interestingly, although both strains showed more than 100 differentially expressed genes (DEGs; FDR < 0.05, absolute $\log_2$-fold change [FC] > 0.5), the number of affected genes and the magnitude of expression changes varied. We identified 267 DEGs in the FY4 *dld3*Δ strain, whereas 744 DEGs were found in the BY4741 *dld3*Δ strain (Tables S1 and S2). Moreover, the average $\log_2$-FC was higher in the BY4741 compared with the FY4 background, with an unexpected bias towards more down-regulated genes in the BY4741 background (Fig 1B and C). Finally, also the identity of the DEGs was different, with only 58 DEGs shared between the two strains (Fig 1D), suggesting that the differential regulation depends on the transcriptional state of the genes rather than the function of the encoded gene products. Consistently, when the directionalities of the expression changes for all genes in the two knockout strains were compared, there was no correlation observed (Fig S1C). Indeed, based on a principal component analysis (PCA) on the gene expression data from the wild-type and *dld3*Δ strains in the two different genetic backgrounds, 94% of the variance was explained by the genetic background rather than the gene-specific deletion (Fig S1D).

## 2-HG alters H3K4 and H3K36 methylation at specific loci

2-HG has previously been shown to increase bulk level methylation of H3K4 and H3K36, but not of H3K79, in yeast (Janke et al, 2017). To test whether 2-HG can alter this methylation also at chromatin-associated histones and evaluate how this effect is distributed across the genome, we performed chromatin immunoprecipitation followed by high-throughput sequencing (ChIP-seq) using the BY4741 wild-type strain and the corresponding *dld3*Δ strain, and antibodies against H3K4me3, H3K36me2, and H3K36me3. The median signal across all genes showed the expected profiles, with H3K36me2/3 enriched across gene bodies and H3K4me3 enriched especially at the transcription start sites (TSSs) and at the end of genes towards the start of the next TSS (Fig 2A). Moreover, the inhibition of histone demethylases, caused by 2-HG accumulation in the *dld3*Δ strain, led to an overall increase in H3K4me3 and H3K36me3, in parallel with a decrease in H3K36me2, when analysing the median signal across all genes.

To obtain a more detailed view of the potentially affected loci, we performed differential peak identification for all three histone modifications using MACS2 and DiffBind (Zhang et al, 2008; Ross-Innes et al, 2012), and the peaks were assigned to genes based on their proximity to the gene locus. This analysis revealed loci with significantly (FDR < 0.05) altered levels of methylation for each modification, when considering methylation changes of at least 0.3-fold on the $\log_2$-scale (Fig 2B–D). For H3K4me3, a comparable number of differentially methylated genes (DMGs) were identified for both increased methylation (157 genes) and decreased methylation (127 genes) (Fig 2B). Interestingly, for both H3K36me2

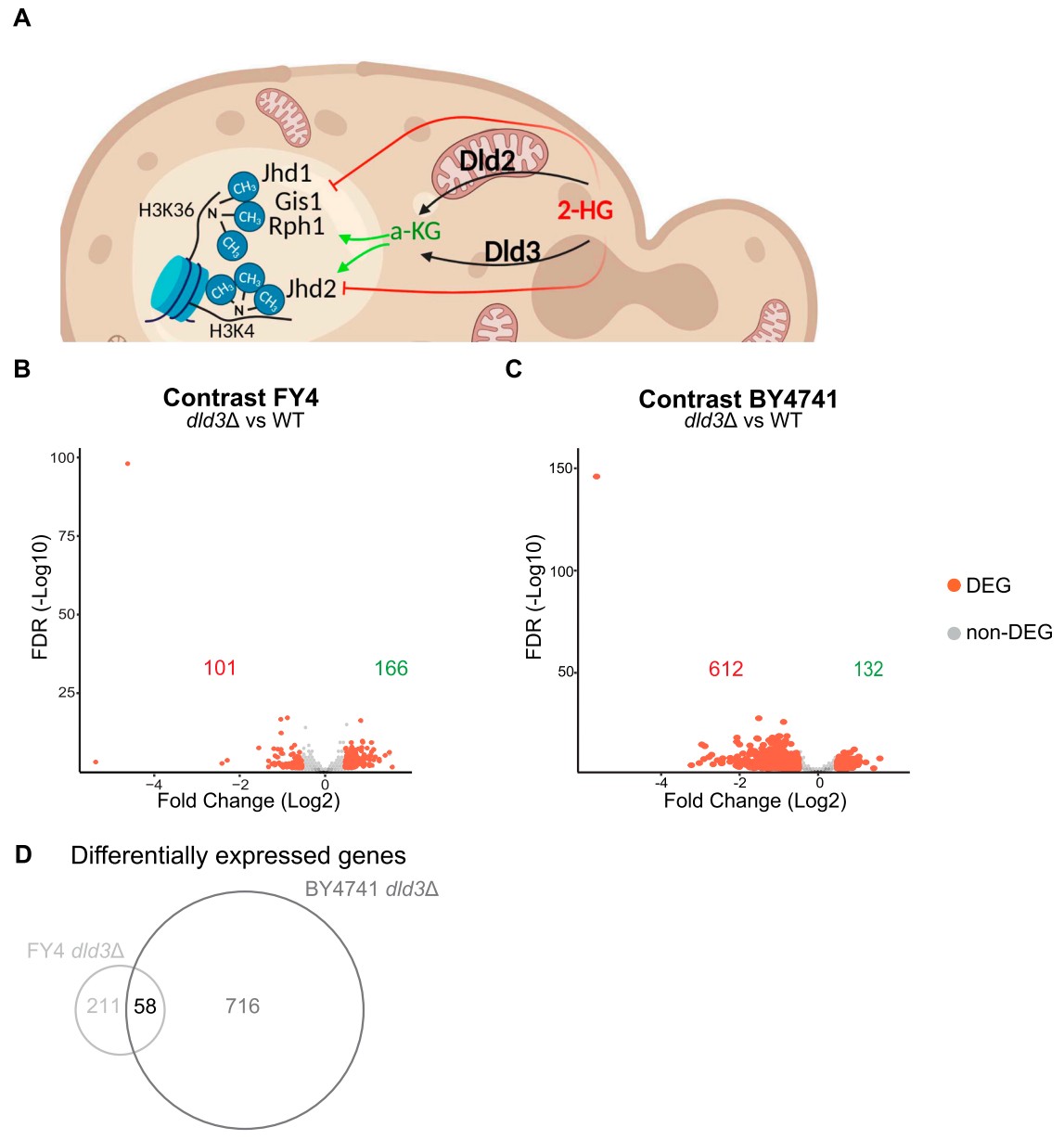

**Figure 1. 2-HG accumulation induces gene expression changes in a genetic background-dependent manner.**
**(A)** Schematic representation (generated with Biorender www.biorender.com) of the role of Dld2 and Dld3 in 2-HG metabolism in yeast and the histone demethylases potentially targeted by 2-HG. **(B, C)** Volcano plots of differential gene expression upon *DLD3* deletion in the FY4 background (B) and the BY4741 background (C). Each dot represents a gene and red dots represent DEGs (FDR < 0.05, absolute[$\log_2$FC] > 0.5). The numbers of down-regulated and up-regulated genes are indicated in red and in green, respectively. **(D)** Venn analysis of DEGs in the FY4 and BY4741 *dld3*Δ strains.

and H3K36me3, hundreds of DMGs with decreased methylation in the *dld3*Δ strain were identified, although only 30 and 21 DMGs, respectively, showed a strong and significant increase (Fig 2C and D). This is in contrast with the increase detected for global H3K36me3 signal based on the metageneplots in Fig 2A. As shown in the examples in Fig 2E, the methylation changes are occurring both in loci where no methylation was present for the wild-type strain, and in loci where an accumulation of methylation is observed in regions where methylation was already present in the wild type. The loci with methylation changes for the three tested

modifications were largely not overlapping with each other (Fig 2F). Interestingly, even though the levels of H3K4me3 and H3K36me3 were generally correlated with the gene expression levels (higher methylation at the genes with higher expression) (Fig S1E–H), the observed locus-specific changes in histone methylation were not followed by changes in the expression of the respective genes (Fig 2G). Specifically, only 9 out of 127 and 1 out of 21 genes with significant increase in H3K4me3 or H3K36me3, respectively, showed a significant increase in gene expression in the *dld3*Δ strain.

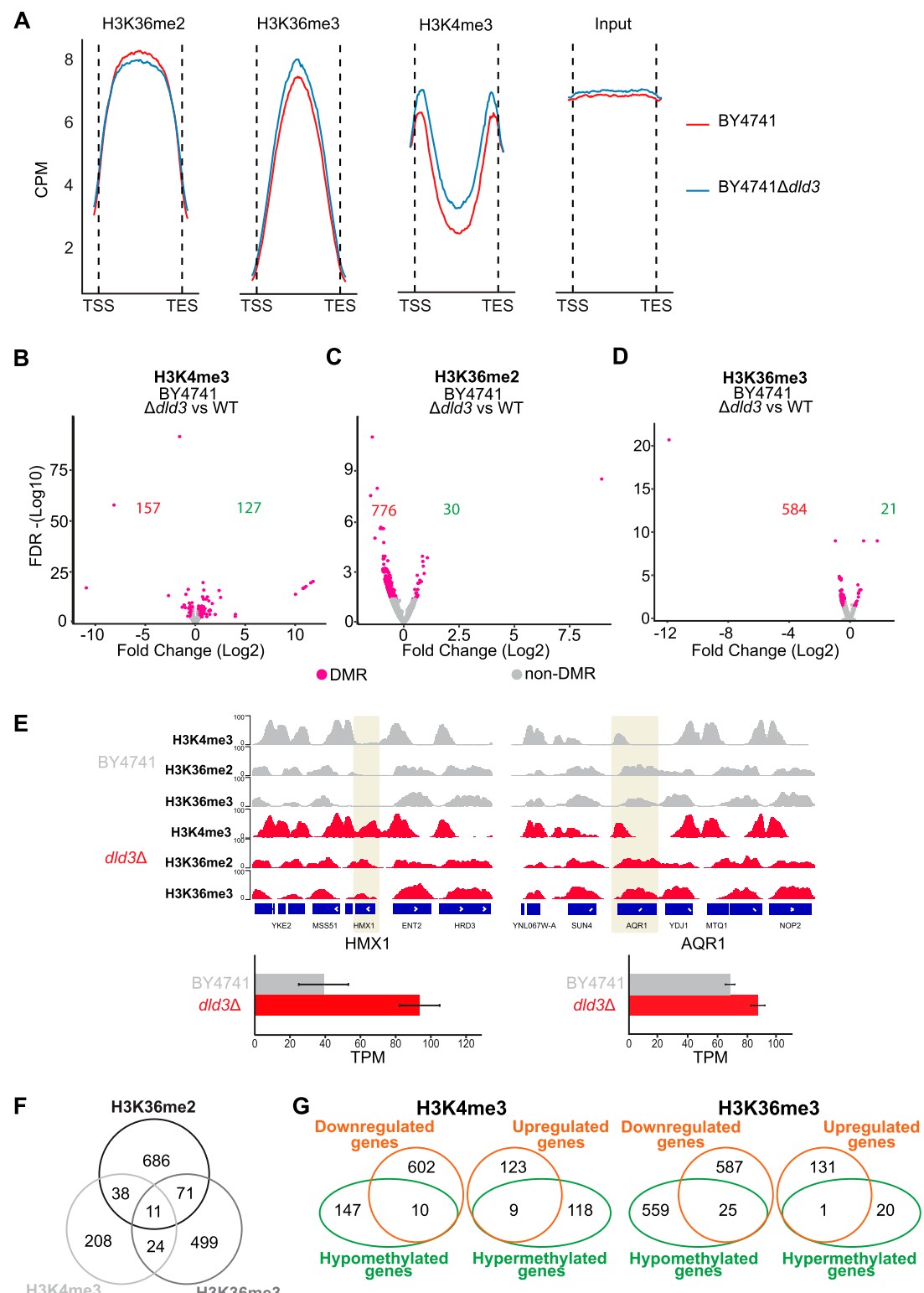

**Figure 2. 2-HG accumulation induces locus-specific changes in H3K36 and H3K4 methylation.**
**(A)** Metagene plots depict the binned median signal from transcription start sites to transcription end sites of all genes for H3K36me2, H3K36me3, H3K4me3, and input from BY4741 WT and BY4741 *dld3*Δ strains. Increase in the H3K36me3 and H3K4me3 methylation is observed upon accumulation of 2-HG, whereas a decrease is observed for the H3K36me2 signal. **(B, C, D)** Volcano plots of differentially methylated regions upon *DLD3* deletion for (B) H3K4me3, (C) H3K36me2, and (D) H3K36me3 in the BY4741 background. Each dot represents a called consensus peak and red dots represent differentially methylated regions (FDR < 0.05, absolute[log₂FC] > 0.3). The numbers of significantly hypomethylated and hypermethylated genes in the *dld3*Δ strain are indicated in red and in green, respectively. **(E)** Visualised example of locus specific

Taken together, these data suggest that 2-HG accumulation can alter both H3K4 and H3K36 methylation, but often at specific and unrelated loci, raising the question whether inhibition of a single, or rather multiple, histone demethylases could explain the observed changes.

## 2-HG induces histone methylation changes via inhibition of several histone demethylases

To better understand the contribution of individual demethylases on the 2-HG-induced methylation changes, we performed ChIP-seq experiments in BY4741 strains deleted in single demethylase genes and in vitro activity assays of the corresponding recombinant demethylases in the absence or presence of 2-HG (Figs 3–5).

First, we profiled H3K4me3 upon deletion of Jhd2, the known yeast H3K4me3 demethylase (Tu et al, 2007), and two of the prominent H3K36 demethylases, Gis1 and Rph1. As expected, the depletion of Jhd2 led to a noticeable increase in median H3K4me3 signal across all genes when compared with the wild-type strain (Fig S2B). Examples of loci with altered H3K4me3 levels (*PHM8* and *ACA1*) are shown in Fig 3A. Interestingly, all three gene deletions induced alterations in H3K4me3 levels, with 95 and 194 DMGs upon deletion of *RPH1* or *GIS1*, respectively (Fig 3B). However, as expected, it was the deletion of *JHD2* that induced by far the biggest changes with over 2,000 DMGs detected (Fig 3B). To compare these H3K4me3 changes with those induced by 2-HG, we performed correlation analysis of the changes at the level of differential peaks between the demethylase deletion strains and the *dld3Δ* strain (Fig 3C–E). The strongest correlation was seen with the *jhd2Δ* strain (R = 0.6), indicating a role for Jhd2 inhibition in 2HG-induced H3K4me3 changes (Fig 3E). The *gis1Δ* strain showed no correlation with *dld3Δ* (R = 0.0051), whereas *rph1Δ* also showed a positive correlation (R = 0.55) (Fig 3C and D). To confirm the ability of 2-HG to directly inhibit Jhd2, we performed in vitro demethylation assays using purified recombinant Jhd2 enzyme and H3K4me3 peptide as substrate, in the presence of increasing 2-HG concentrations (Fig 3F). These experiments showed that 2-HG inhibits the demethylase activity of Jhd2 significantly already at concentrations of 0.5 mM and above, suggesting that Jhd2 inhibition by 2-HG may account for increased H3K4me3 levels observed in the *dld3Δ* strain. Consistently, an estimation of half maximal inhibitory concentration (IC50) for Jhd2 from these in vitro data resulted in the concentration of 2.2 mM.

Next, we focused on H3K36me2 and profiled its changes in deletion mutants for all four yeast demethylases. An example of a locus with altered H3K36me2 (*RNR3*) is shown in Fig 4A. From 100 up to 1,600 H3K36me2 DMGs could be detected in each mutant, with least changes observed upon *JHD2* deletion, consistent with its role in regulating H3K4me3 rather than H3K36me2 levels (Fig 4B). Surprisingly, but consistent with the results obtained in the *dld3Δ* strain,

most loci significantly affected by deletion of the H3K36 demethylases showed decreased–rather than increased–H3K36me2 signal, with the biggest changes observed in the *rph1Δ* strain (Fig 4B), although no changes were observed for the global median H3K36me2 signal (Fig S2C). Moreover, a comparison to *dld3Δ* revealed a positive correlation with both the *rph1Δ* (R = 0.58) and *jhd1Δ* (R = 0.76) strains, whereas to a lesser extent for *gis1Δ* (R = 0.44) and *jhd2Δ* (R = 0.42) (Fig 4C–F). Particularly, the *gis1Δ* strain, despite having a greater number of DMGs (464) compared with *jhd1Δ* (223), had a lower Pearson correlation. An in vitro analysis with H3K36me2 peptide showed that 2-HG exerts an inhibitory effect on all three H3K36 demethylases with an estimated IC50 of 3.1 mM and 2.0 mM for Gis1 and Jhd1, respectively. However, Rph1 was most sensitive to this effect under the assay conditions used, with significant inhibition observed starting from 1 mM 2-HG and an estimated IC50 of 1.3 mM (Fig 4G). Based on this, the high number of affected genes, and the high correlation of H3K36me2 changes between *rph1Δ* and *dld3Δ*, 2-HG induced H3K36 methylation changes may be preferentially, albeit not solely, mediated through Rph1 inhibition.

To test whether this could also be the case for H3K36me3, we profiled this modification in three demethylase knockout strains (*rph1Δ*, *gis1Δ*, and *jhd2Δ*), including those known to demethylate H3K36me3. Examples of loci with altered H3K36me3 levels (*YRO2* and *GDB1*) are shown in Fig 5A. Here, *RPH1* deletion clearly induced the most profound changes, with hundreds of DMGs and most genes showing decreased methylation (Fig 5B). This is again in contrast with the changes in the median H3K36me3 signal across all genes (Fig S2C), similarly to what was observed for *dld3Δ* cells (Fig 2). Moreover, *gis1Δ* showed almost no DMGs for H3K36me3 and for *jhd2Δ* mostly increased methylation at a limited number of loci could be detected (Fig 5B). A particularly strong positive correlation was observed between the 2-HG-accumulating *dld3Δ* strain and the *rph1Δ* strain when focusing on H3K36me3 (R = 0.61) (Fig 5C), with no correlation for the other two deletion strains (Fig 5D and E), suggesting that 2-HG induced H3K36me3 changes may be mostly mediated by Rph1 inhibition. Indeed, in vitro demethylation assays using the H3K36me3 peptide confirmed a relatively potent inhibitory effect of 2-HG on Rph1 activity under the assay conditions used, with an estimated IC50 value of 1.5 mM, compared with IC50 of 5.0 mM for Gis1 (Fig 5F).

Taken together, our analyses show that all tested demethylases (Rph1, Jhd1, Jhd2, Gis1) are, at various degrees, inhibited by 2-HG, and that the inhibition of all these demethylases is likely to contribute to histone methylation changes observed upon intracellular 2-HG accumulation. However, the most potent inhibitory effect by 2-HG was observed with Rph1 and this is in good agreement with H3K36 methylation being most profoundly affected by *RPH1* deletion that showed good correlation with the *DLD3* deletion strain. It should be noted that our in vitro demethylase

changes upon accumulation of 2-HG in the BY4741 background. For the *HMX1* gene, we can see the formation of peaks for all the studied modifications in the *dld3Δ* compared with the WT strain. The changes in methylation are accompanied by increased gene expression (represented in TPM). For the *AQR1* gene, we can see an increase in the H3K4me3 and H3K36me3 signal upon 2-HG accumulation, whereas no change is observed for the H3K36me2 modification. A change in gene expression is also observed. **(F)** Overlap between differentially methylated genes (DMGs) (FDR < 0.05, absolute[log$_2$FC] > 0.3) for the analysed histone modifications. Venn analysis of DMGs shows weak coupling between methylation changes in different histone residues. **(G)** Overlap between the DMGs and the DEGs. For H3K4me3, only nine genes were overexpressed upon increase in H3K4me3 after 2-HG accumulation (*dld3Δ* strain). Similarly, for the H3K36me3 modification, the increase in methylation was accompanied by a significant increase in expression for only one gene.

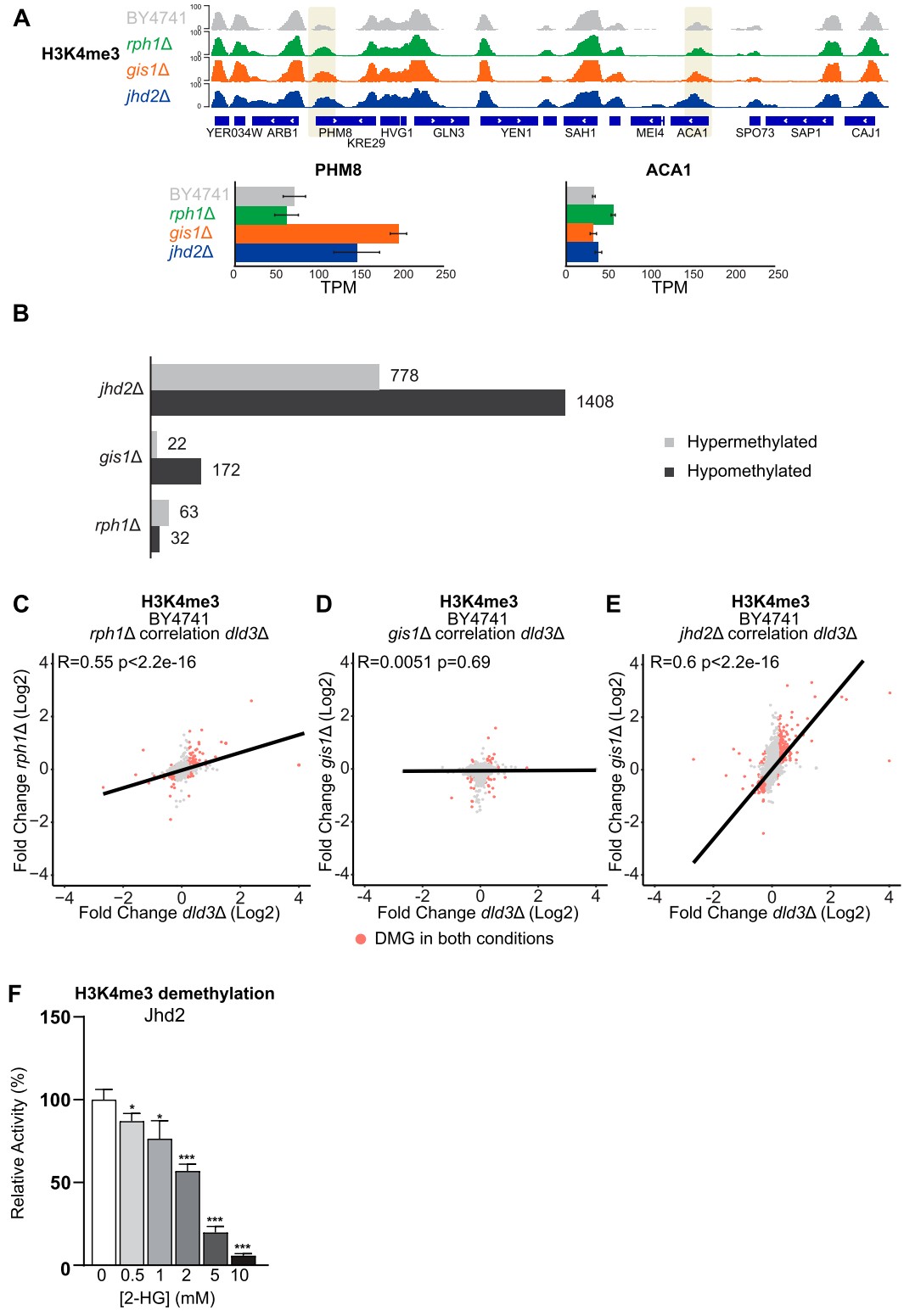

**Figure 3. Locus-specific changes in H3K4me3 and inhibition of Jhd2 by 2-HG.**
**(A)** Locus-specific changes for H3K4me3 in the *gis1Δ*, *jhd2Δ*, and *rph1Δ* strains shown for the *PHM8* and *ACA1* genes. *PHM8* shows increased H3K4me3 in all tested mutant strains, but only in *gis1Δ* and *jhd2Δ*, the changes are coupled with gene expression changes. At *ACA1*, H3K4me3 increases in *jhd2Δ* strain, but no significant gene expression change could be detected in this strain. **(B)** Number of DMGs (FDR < 0.05, absolute[$\log_2$FC] > 0.3) for H3K4me3 in the *jhd2Δ*, *gis1Δ*, and *rph1Δ* strains. Grey bars represent hypermethylated genes, whereas black bars represent hypomethylated genes. The strongest changes are observed for the *jhd2Δ* strain with 2186 DMGs. Most of these genes are hypomethylated. **(C, D, E)** Correlation analysis between differentially methylated genes in different histone demethylase deletion strains and the *dld3Δ* strain.

assays were performed in the presence of methylated peptide and a-KG concentrations (1 mM) that are likely to saturate the enzymes (Krishnan et al, 2012). Even stronger inhibitory effects by 2-HG may be expected in the presence of lower, more physiological concentrations (<0.5 mM; Bennett et al, 2009; Becker-Kettern et al, 2016) of the a-KG cosubstrate.

## 2-HG-induced gene expression changes are consistent with preferential inhibition of Rph1

Having confirmed that 2-HG can inhibit several of the yeast histone demethylases with consequences on locus-specific histone methylation profiles, we asked whether all of them can also contribute to the 2-HG-induced gene expression changes. With this goal in mind, we performed transcriptomic profiling of the *rph1Δ*, *gis1Δ*, *jhd1Δ*, and *jhd2Δ* strains in comparison with the wild-type BY4741 strain (Fig 6). Deletion of each of the demethylases induced changes in the expression of >100 genes with strongest changes observed upon *RPH1* deletion (2,446 DEGs) and *JHD1* deletion (938 DEGs) (Fig 6A–D and Table S3). Interestingly, whereas *gis1Δ* and *jhd1Δ* both showed comparable numbers of up- and down-regulated genes (Fig 6B and C), the directionality of changes was skewed for both *rph1Δ* and *jhd2Δ*. Specifically, the deletion of *JHD2* led to more up- than down-regulated genes, whereas *RPH1* deletion induced higher gene silencing (Fig 6A and D). This distribution of changes was reminiscent of the changes in H3K36me2 and H3K36me3 profiles in the same mutant strains (Figs 4 and 5), suggesting that H3K36 methylation, rather than H3K4 methylation, is linked to the 2-HG-induced gene expression changes. Moreover, the magnitude and directionality of the expression changes upon *RPH1* deletion were surprisingly similar to those observed in the *DLD3* deletion strain (Fig 1C).

To directly address the similarity of the presumably 2-HG-induced gene expression changes in the *dld3Δ* strain to those in the individual histone demethylase mutants, we performed further correlation analyses between the datasets (Fig 6E–H). Strikingly, *rph1Δ* showed a clear positive correlation (R = 0.71) with *dld3Δ* at the gene expression level (Fig 6E), whereas no correlation with the other demethylase deletion mutants was observed (Fig 6F–H), further supporting Rph1 as the main mediator of gene expression changes induced by 2-HG accumulation, possibly because of its preferential inhibition by the latter.

Finally, to investigate all samples together in an unsupervised manner, we performed PCA of the transcriptome profiles of all the BY4741 deletion mutants and the respective wild-type control strain (Fig 6I). Consistent with the largest gene expression changes, *rph1Δ* strain clustered furthest from the wild-type control and other strains along PC1, explaining up to 61% of the variance between the strains. The *jhd2Δ* and *gis1Δ* strains clustered largely together with the wild-type strain, whereas *jhd1Δ* moved further away along PC2.

Importantly, the transcriptome of the *dld3Δ* strain was shifted away from the other strains along PC1 towards *rph1Δ*, further supporting the similarity of changes induced by *RPH1* and *DLD3* deletion.

Taken together, although deletion of all histone demethylases induced gene expression changes, only the changes induced by *RPH1* deletion correlated with those induced by *DLD3* deletion, in keeping with preferential inhibition of Rph1 by 2-HG.

## Rph1 inhibition down-regulates repressed genes marked by high H3K36me2 and low H3K36me3

Although Rph1 appeared to be preferentially inhibited by 2-HG and to explain a large part of methylation changes and most of the gene expression changes induced by 2-HG accumulation, it remained unclear how Rph1 inhibition leads to these changes. Indeed, most of the DMGs, in both the *dld3Δ* and *rph1Δ* strains, were not associated with differential gene expression (Figs 2G and S2). Moreover, most of the DMGs and DEGs showed decreased methylation for H3K36 and down-regulation of gene expression, respectively (Figs 1, 2, 5, and 6), whereas the median H3K36me3 signal across all genes was increased in both deletion strains. To have a more detailed look at the H3K36 methylation profiles at the DEGs, we analysed metagene plots of different subgroups of genes for all the used BY4741 strains. When considering all genes, a maximum median signal of ~8 counts per million (CPM) could be observed (Figs 2A and S2). When focusing specifically on DEGs up-regulated in the *dld3Δ* strain, only minor differences between the strains in either H3K36me2 or H3K36me3 could be observed, whereas the overall H3K36me3 signal was elevated (median CPM >10) when compared with all genes (Fig 7A). However, when plotting the median signals for ~600 genes down-regulated in the *dld3Δ* strain, the median signal of H3K36me3, but not that of H3K36me2, showed an increased trend in the *dld3Δ* and *rph1* strains compared with other strains. And when focusing on a larger but overlapping group of ~1,400 genes down-regulated upon *RPH1* deletion, this difference became more pronounced in both the *rph1Δ* and *dld3Δ* strains (Fig 7B), suggesting that down-regulation of genes upon *RPH1* inhibition or deletion was associated with increased H3K36me3 levels.

This result was puzzling as increased levels of H3K36me3 have typically been associated with high levels of transcription (Tippmann et al, 2012; Becker et al, 2019). However, the overall signal of H3K36me3 at the down-regulated genes was particularly low, with a median signal of only 3–5 CPM, compared with 7.5–12 CPM for up-regulated genes (Fig 7A and B). At the same time, H3K36me2 signals remained comparably high at both up- and down-regulated genes. This difference became even more striking when focusing on the top 500 most down-regulated genes in either the *dld3Δ* or *rph1Δ* strain, with the median H3K36me3 signal reaching only approximately 2 CPM (Fig 7C), a signal that would not be detected by most peak calling tools. This suggests that the genes with the strongest

---

**(C, D, E)** Log$_2$-FCs in H3K4me3 for the Δ*dld3* strain are plotted on the x-axis, whereas log$_2$-FCs in the (C) *rph1Δ*, (D) *gis1Δ*, and (E) *jhd2Δ* mutant strains are plotted on the y-axis. Each dot represents a gene and red dots represent genes found to change significantly for H3K4me3 in both mutant strains (FDR < 0.05, absolute[log$_2$FC] > 0.3). Pearson correlation is calculated for each set. The strongest correlation to the *dld3Δ* strain is observed for the *rph1Δ* and *jhd2Δ* strains. **(F)** Inhibition of Jhd2 enzymatic activity by 2-HG. A demethylation assay using recombinant Jhd2 and an H3K4me3 peptide as substrate was performed in vitro in the presence of increasing 2-HG concentrations. Enzymatic activities were determined based on initial velocities (in $\mu$M NADH.min$^{-1}$) and normalized to the value obtained in the absence of 2-HG (set to 100%). Data shown are means ± SDs from three independent experiments. Statistical significance was calculated using $t$ test; * = $P$ < 0.05, ** = $P$ < 0.01 *** = $P$ < 0.001.

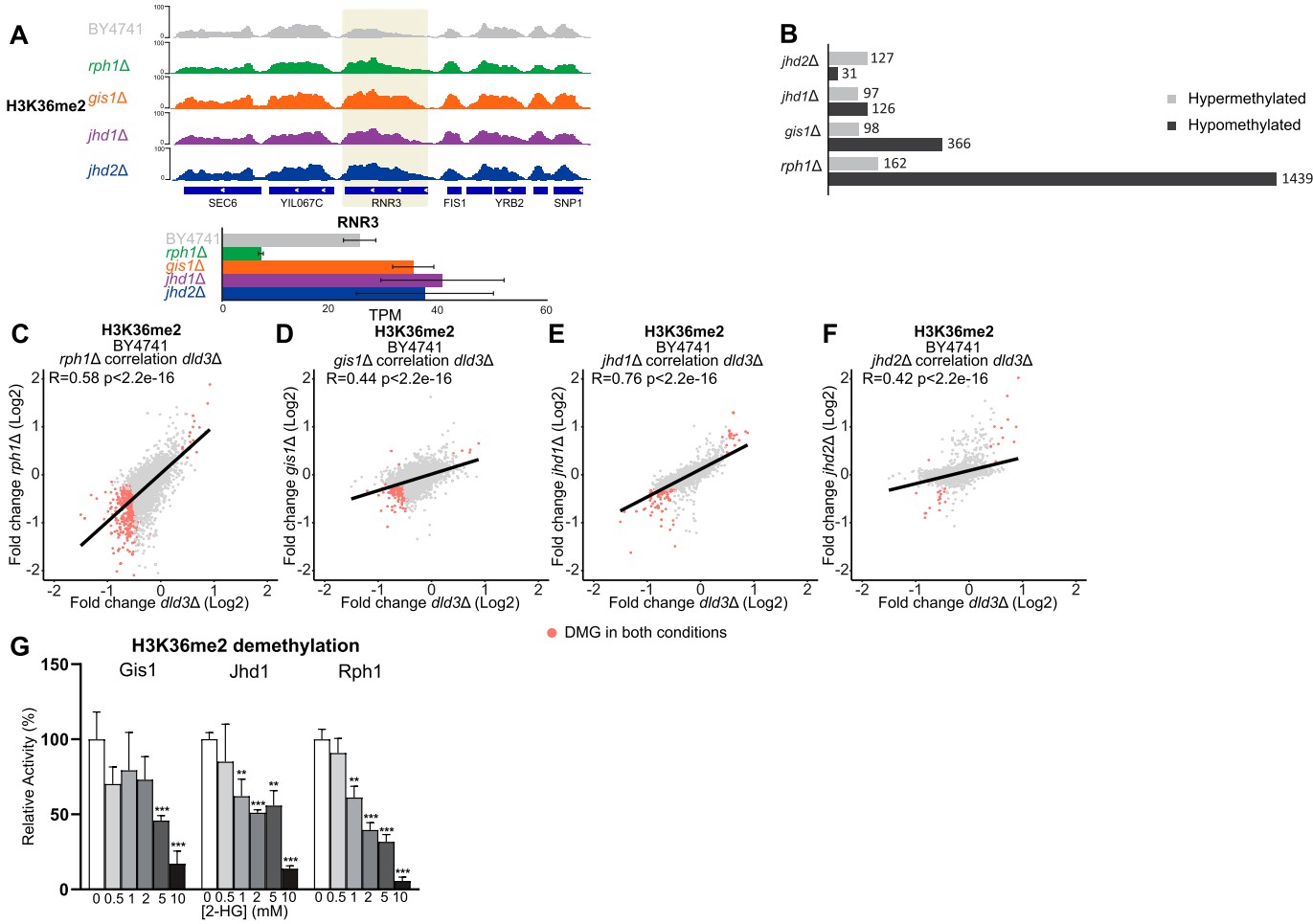

**Figure 4.   Locus-specific changes in H3K36me2 and inhibition of Gis1, Jhd1, and Rph1 by 2-HG.**
**(A)** Locus-specific changes for H3K36me2 in the *rph1Δ*, *gis1Δ*, *jhd1Δ*, and jhd2Δ strains shown for the *RNR3* gene. **(B)** Number of DMGs (FDR < 0.05, absolute[log₂FC] > 0.3) with differential H3K36me2 in *rph1Δ*, *gis1Δ*, *jhd1Δ*, and *jhd2Δ* are shown as in Fig 3B. The greatest number of DMGs was observed for the *rph1Δ* strain. **(C, D, E, F)** Correlation analysis between differentially methylated genes in different histone demethylase deletion strains and the *dld3Δ* strain. **(C, D, E, F)** Log₂-FCs of H3K36me2 in the *dld3Δ* strain are plotted on the x-axis, whereas log₂-FCs in the (C) *rph1Δ*, (D) *gis1Δ*, (E) *jhd1Δ*, and (F) *jhd2Δ* strains are plotted on y-axis. Each dot represents a gene and red dots represent genes found to change significantly for H3K36me2 in both mutant strains (FDR < 0.05, absolute[log₂FC] > 0.3). Pearson correlation is calculated for each set. The strongest correlation with the *dld3Δ* strain is observed for the *rph1Δ* and *jhd1Δ* strains. **(G)** Inhibition of the Gis1, Jhd1, and Rph1 enzymatic activities by 2-HG. A demethylation assay using recombinant enzymes and an H3K36me2 peptide as substrate was performed in vitro in the presence of increasing 2-HG concentrations. Enzymatic activities were determined based on initial velocities (in $\mu$M NADH.min$^{-1}$) and normalized to the value obtained in the absence of 2-HG (set to 100%). Data shown are means ± SDs from three independent experiments. Statistical significance was calculated using *t* test; * = *P* < 0.05, ** = *P* < 0.01, *** = *P* < 0.001.

down-regulation upon Rph1 inhibition could also be expressed at low levels already at baseline (Fig 7D). Indeed, many of the genes down-regulated upon 2-HG accumulation showed high H3K36me2 but were devoid of H3K36me3 and H3K4me3 in the wild-type BY4741 control strain with some increase in H3K36me3 observed in *dld3Δ* and *rph1Δ* strains, as illustrated by the example of the *PCH2* and *HXT5* genes in Fig 7E. Moreover, comparison of the basal gene expression levels between strains confirmed a low basal expression of the genes down-regulated upon *RPH1* and *DLD3* deletion (Fig 7D). Importantly, the lower basal expression of these genes was detectable also in the wild-type strain and the other strains where these genes were not down-regulated, therefore indicating that 2-HG-mediated inhibition of Rph1 can lead to further repression of already silenced or low expressed genes, possibly through increased H3K36me3 at loci marked by high H3K36me2 and

absence of H3K4me3. This is consistent with the enriched gene ontology (GO) categories among the down-regulated genes, many of which are linked to meiotic cell cycle, sporulation, and reproductive processes that should be silenced in the haploid BY4741 strain (Table S4). Thus, inhibition of Rph1 appears to be the main mediator of the extensive down-regulation of genes induced by 2-HG, possibly because of increased H3K36me3, especially at repressed loci enriched for H3K36me2 and devoid of H3K4me3.

# Discussion

Accumulation of 2-HG in human cells has been shown to drive tumorigenic gene expression, disrupt DNA repair, and inhibit cellular differentiation (Ye et al, 2018; Pirozzi & Yan, 2021). Albeit 2-HG could

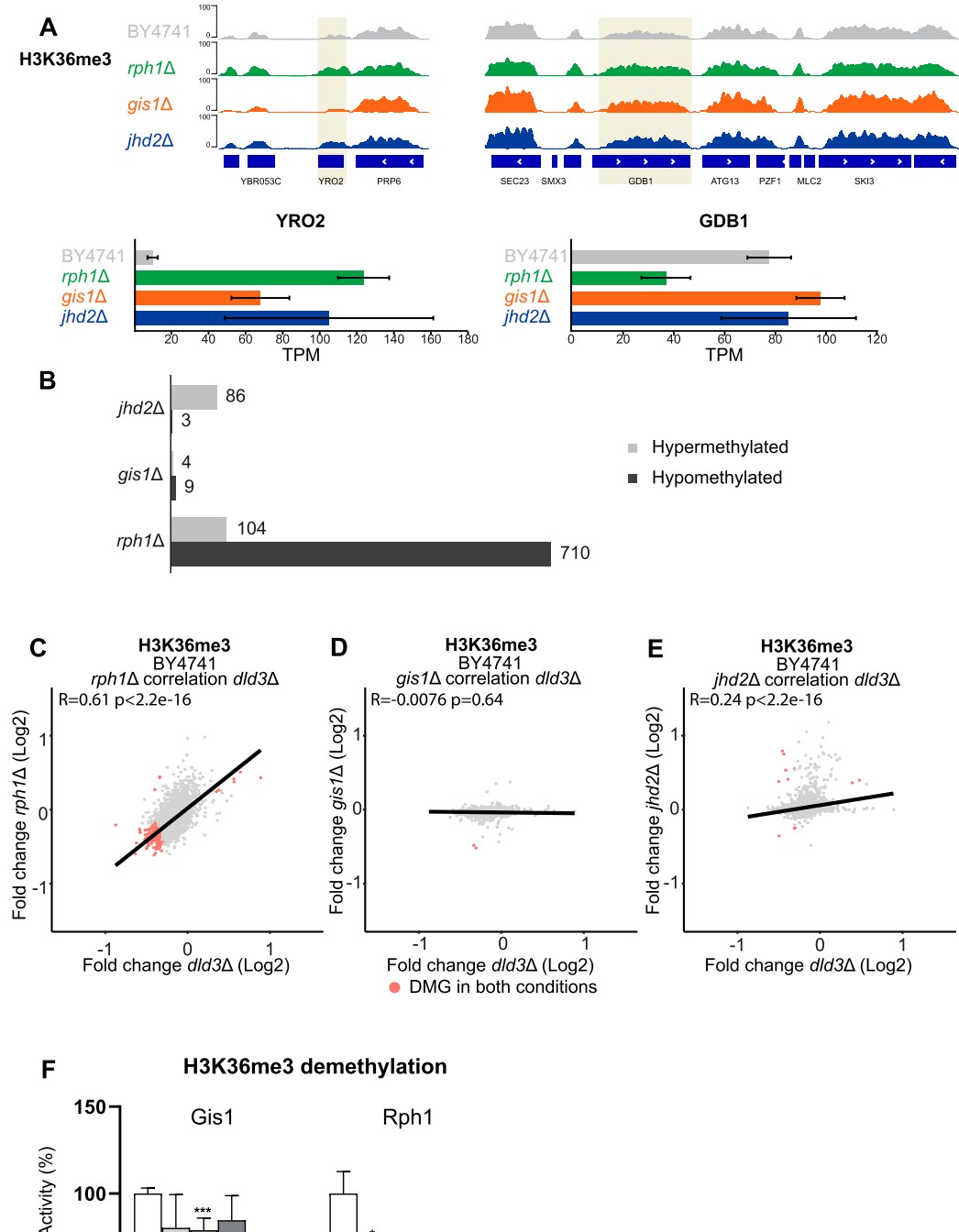

**Figure 5. Locus-specific changes in H3K36me3 and inhibition of Gis1 and Rph1 by 2-HG.**
**(A)** Locus-specific increase for H3K36me3 in the *rph1Δ* and *gis1Δ* strains shown for the *YRO2* and *GDB1* genes. Increased methylation at *YRO2* is accompanied by changes in gene expression while GDB1 is repressed specifically in *rph1Δ* strain. **(B)** The number of DMGs (FDR < 0.05, log2FC > 0.3) for H3K36me3 in the *rph1Δ*, *gis1Δ*, and *jhd2Δ* strains is shown as in Fig 3B. **(C, D, E)** Correlation analysis between differentially methylated genes in different histone demethylase deletion strains and the *dld3Δ* strain. **(C, D, E)** Log$_2$-FCs in H3K36me3 in the *dld3Δ* strain are plotted on the x-axis, whereas log$_2$-FCs in the (C) *rph1Δ*, (D) *gis1Δ*, and (E) *jhd2Δ* mutant strains are plotted on the y-axis. Each dot represents a gene and red dots represent DMGs found to change significantly for H3K36me3 in both mutant strains. Pearson correlation is calculated for

potentially inhibit up to 70 a-KG-dependent enzymes, disruption of DNA and H3K9 demethylation via inhibition of TET family deme- thylases and histone lysine demethylases such as KDM4, respectively, have been identified as main mediators of the cancer-associated phenotypes (Losman et al, 2020). The proposed mechanisms un- derlying the gene expression changes and consequent cellular phenotypes include DNA methylation-induced disruption of insu- lator regions required for proper topological domain formation and impaired accessibility of key genes controlling differentiation be- cause of increased H3K9 methylation. However, many other histone modifications, including H3K4 and H3K36 methylation, are known to be affected by high 2-HG levels (Lu et al, 2012; Turcan et al, 2018). It has remained unclear whether changes in these euchromatin marks can contribute to the 2-HG-induced gene expression changes or whether those effects are masked by changes induced by alterations in re- pressive heterochromatin.

Here, we aimed at investigating the impact of 2-HG on gene ex- pression in *S. cerevisiae*, a model organism devoid of repressive chromatin marks (lack of DNA methylation and histone methylation at residues such as H3K9). In comparison with human cells, only four demethylases (Rph1, Gis1, Jhd1, and Jhd2) targeting histone lysine residues are known to be present in *S. cerevisiae*, all belonging to the JmjC domain-containing family of proteins. Interestingly, we did observe global and locus-specific changes in histone methylation and in gene expression in yeast strains deficient in the main enzyme responsible for 2-HG degradation in this species, namely Dld3 (Figs 1 and 2). Disruption of the latter results in high intracellular 2-HG accumulation as a main consequence, suggesting that the histone methylation and gene expression changes observed in *dld3*Δ strains are largely mediated by the high 2-HG levels. However, the changes in histone methylation and gene expression often affected different loci and the changes in gene expression were found to depend on the genetic background of the yeast strain. Importantly, although all four yeast demethylases were inhibited at least to some extent by 2- HG, we could show that Rph1 is most sensitive to this effect and, concomitantly, its deficiency led to the biggest changes in histone methylation and induced gene expression changes that correlated most closely with the ones observed upon *DLD3* deletion.

In this study, we have used genetic knockout strains that have adapted to the loss of the deleted enzymes. Because histone methylation is a balance of the activities of methyltransferases and demethylases, our results do not necessarily reflect the primary changes occurring upon inhibition or deletion of demethylases but rather the acquired steady states. Observing the median methyl- ation signals across all genes for 2-HG accumulating *dld3*Δ cells revealed an expected change of increased H3K4me3 and H3K36me3 at the expense of reduced H3K36me2 (Fig 2A) Similarly, deletion of specific histone demethylases led to an anticipated accumulation of methylation across genes (Fig S2). However, a statistical iden- tification of DMGs with DiffBind revealed a different picture with variable responses at individual loci (Fig 2B). This apparent

contradiction can be partially explained by the procedure of DMG identification that requires the presence of a sufficient signal to be called as a peak by MACS2, biasing the analysis to loci with already detectable methylation, whereas an increase in methylation was observed in particular at down-regulated genes with low levels of H3K4 or H3K36 methylation (see the Materials and Methods section for details). In addition to this technical explanation, other bio- logical mechanisms could also explain why more of the already methylated loci showed reduced, rather than increased methyla- tion upon the deletion of the corresponding demethylase (Figs 3–5).

One possible explanation could be the simultaneous down- regulation of histone methyltransferases in response to histone demethylase deletion. However, none of the relevant methyl- transferases (Set1, Set2, and Dot1) or the members of the COMPASS complex changed in their mRNA expression in any of the histone demethylase deletion strains (Tables S1 and S3). Nevertheless, we cannot exclude changes in protein abundance or localization that could affect the compensation of loss of demethylation differently at different loci (Soares et al, 2014).

Other possibilities include an altered availability of the methyl donor or interdependence between methylation and demethyla- tion processes. It has been previously shown that a feedback control exists between the expression of enzymes controlling a-KG availability and demethylase activity (Di Nisio et al, 2023). A similar feedback could also exist for availability of S-adenosylmethionine (SAM) for methylation, although our transcriptome data indicated no changes in the expression of Sam1 or Sam2, enzymes which are responsible for SAM synthesis in yeast (Tables S1 and S3). Still, a feedback between demethylase and methyltransferase activities and co-regulation of their target genes have been reported before for H3K4me3 (Ramakrishnan et al, 2016; Di Nisio et al, 2023). In detail, a loss of either the methyltranferase or the demethylase can lead to comparable changes in the target gene expression, depending on the chromatin context, although opposing outcomes would be expected. Such explanation might exist for differential responses to loss of demethylase activity also for other targets like H3K36me3. Lastly, it has been shown that yeast cells can regulate histone methylation levels and respond to stimuli, such as carbon source availability, through DNA replication and cell cycle (Sein et al, 2015). It could be that our yeast strains have adapted their methylation levels through several cell cycles.

Our results on locus-specific epigenetic changes upon 2-HG accumulation are consistent with observations from other or- ganisms. For example, overexpression of mutant IDH1 in human astrocytes can induce methylation changes for a number of histone residues, including H3K36me3, only in specific regions, with a few hundred genes showing differential expression (Turcan et al, 2018). Similarly, genome-wide analysis of DNA methylation changes in gliomas has revealed locus-specific hypermethylation that is limited to a few hundred loci (Flavahan et al, 2016). This could be because of selective localization of the inhibited enzymes, or the

---

each set. The strongest correlation was observed between the *dld3*Δ and *rph1*Δ strains. **(F)** Inhibition of the Gis1 and Rph1 enzymatic activities by 2-HG. A demethylation assay using recombinant enzymes and H3K36me3 peptide as substrate was performed in vitro in the presence of increasing 2-HG concentrations. Enzymatic activities were determined based on initial velocities (in $\mu$M NADH.min$^{-1}$) and normalized to the value obtained in the absence of 2-HG (set to 100%). Data shown are means ± SDs from three independent experiments. Statistical significance was calculated using *t* test; * = $P < 0.05$, ** = $P < 0.01$, *** = $P < 0.001$.

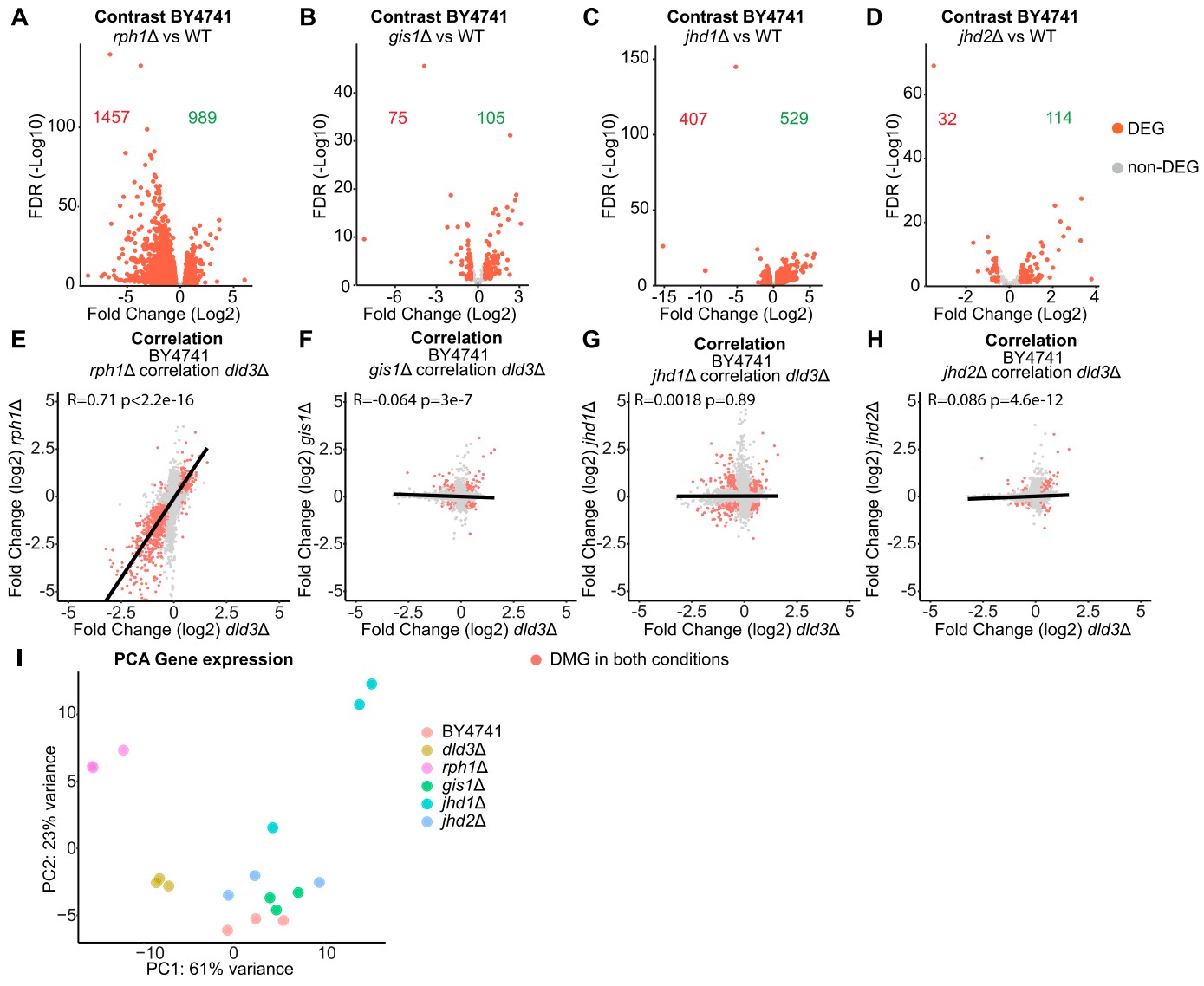

**Figure 6. Transcriptomic analysis of demethylase deletion strains implicates Rph1 as the main mediator of 2-HG–induced gene expression changes.**
**(A, B, C, D)** Volcano plots of differential gene expression upon deletion of (A) *RPH1*, (B) *GIS1*, (C) *JHD1*, and (D) *JHD2*. Each dot represents a gene and red dots represent DEGs (FDR < 0.05, absolute[$\log_2$FC] > 0.5). The number of down-regulated and up-regulated genes is indicated in red and in green, respectively. **(E, F, G, H)** Correlation analysis between DEGs in different histone demethylase deletion strains and the *dld3Δ* strain. **(E, F, G, H)** Gene expression $\log_2$FCs in the *dld3Δ* strain are plotted on the x-axis, whereas $\log_2$FCs in the (E) *rph1Δ*, (F) *gis1Δ*, (G) *jhd1Δ*, and (H) *jhd2Δ* mutant strains are plotted on the y-axis. Each dot represents a gene and red dots represent genes found to be significantly differentially expressed in both mutant strains (FDR < 0.05, absolute[$\log_2$FC] > 0.5). Pearson correlation is indicated for each comparison. The strongest correlation was observed between the *rph1Δ* and *dld3Δ* strains. **(I)** Principal component analysis reveals the transcriptomic similarity between the *rph1Δ* and *dld3Δ* strains. Three independent biological replicates of each strain are depicted in the indicated colours.

metabolites involved in enzymatic reactions within the nucleus. Rph1 is known to occupy less than 700 loci in the yeast genome, mostly in transcribed regions of genes (Shu et al, 2020). Alternatively–or in addition–varying expression levels of the targeted demethylases between yeast strains could contribute to the context-dependent differences. Indeed, also in our data, we could observe largely separate cohorts of DEGs between the studied strains, FY4 and BY4741 (Fig 1). Interestingly, Rph1 is expressed at fourfold higher levels in the BY4741 strain than in the FY4 strain, in keeping with the higher magnitude of gene expression and H3K36 methylation changes in the former strain. Finally, differences in metabolism

between the strains (FY4 is a prototrophic strain, whereas BY4741 is auxotrophic for histidine, leucine, methionine, and uracil) could also contribute to the observed differences. The BY4741 strain is phenotypically similar to FY4 when proper media supplementation is provided. However, even if in auxotrophic strains, adequate media supplementation might restore a similar growth phenotype to the corresponding prototrophic strain, transcription differences are still detectable especially for auxotrophies in the *LEU2* and *MET15* genes (Alam et al, 2016). Further analyses are warranted to understand why epigenetic changes induced by 2-HG accumulation differ between FY4 and BY4741, and possibly other strains.

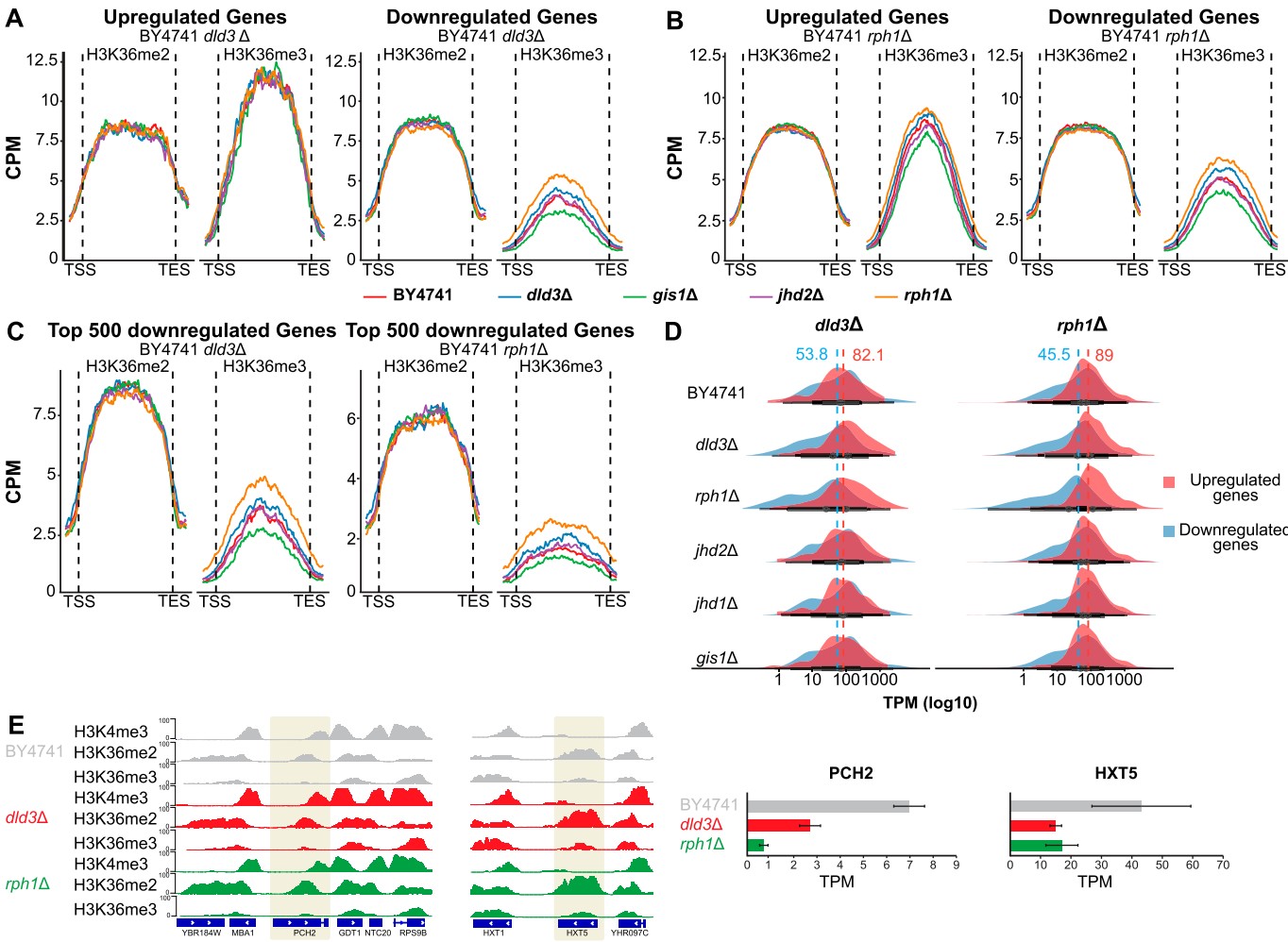

**Figure 7. Rph1 inhibition by 2-HG leads to increased repression of silenced genes.**
**(A, B, C)** Metagene plots depict the binned median signal for H3K36me2 and H3K36me3 from transcription start sites to transcription end sites of genes that are up- or down-regulated upon *DLD3* (A) or *RPH1* deletion (B) (FDR < 0.05, absolute[log$_2$FC] > 0.5), or for the top 500 most down-regulated genes in the two BY4741 deletion strains (C). Red indicates the ChIP-seq signal in wildtype, blue in *dld3Δ*, green in *gis1Δ*, purple in *jhd2Δ*, and orange in *rph1Δ*. **(D)** Basal gene expression distribution (TPM values) of the genes up- or down-regulated (FDR < 0.05, absolute[log$_2$FC] > 0.5) in the *dld3Δ* strain (on the left) or *rph1Δ* strain (on the right). Median expression of these genes across all analysed strains is indicated by the dashed lines, illustrating the low median expression of genes targeted for down-regulation already at baseline. **(E)** *PCH2* represents an example of a gene down-regulated upon *DLD3* and *RPH1* deletion and shows no H3K4me3 methylation, high H3K36me2, and very low H3K36me3 levels (but that are modestly increased in the deletion strains). Similarly, *HXT5* is also down-regulated in the *rph1Δ* and *dld3Δ* strains and associated with comparable chromatin states.

Similarity of the histone methylation and gene expression alterations between the *dld3Δ* and *rph1Δ* strains, together with the in vitro demethylation assays, support a key role for Rph1 in mediating 2-HG-induced epigenetic changes. Interestingly, the closest homologs of Rph1 in humans are members of the KDM4 subfamily of histone demethylases (Allis et al, 2007; Liang et al, 2011), that have also been shown to be preferential targets of 2-HG inhibition (Chowdhury et al, 2011; Laukka et al, 2018; Losman et al, 2020), and that catalyse demethylation of H3K9, the modification linked to many of the 2-HG-induced phenotypes (Lu et al, 2012; Inoue et al, 2016; Juan-Manuel et al, 2019). Moreover, KDM4B inhibition has been shown to inhibit DNA repair through H3K9 hypermethylation (Sulkowski et al, 2020). Therefore, our results support the concept of preferential enzyme inhibition underlying 2-HG-associated phenotypes and can aid the design of interventions focused on these preferential targets to reverse such phenotypes.

In yeast, Rph1 acts both as a histone demethylase and a transcriptional repressor controlling, among others, ribosome production, autophagy, and cell cycle (Bernard et al, 2015; Eapen et al, 2017; Shu et al, 2020). Accordingly, a slower growth phenotype was detectable also for our *rph1Δ* strains, whereas this was not the case for the *dld3Δ* strain (data not shown). However, the different functions of Rph1 in processes such as autophagy and cell cycle are known to be independent of Rph1 histone demethylase activity, providing the most likely explanation for the differences in the phenotypes. Alternatively, inhibition by 2-HG could be incomplete, allowing the cells to escape these phenotypes.

Although 2-HG accumulation did not trigger a detectable growth phenotype under the cultivation conditions used, it was still sufficient to induce extensive repression of over 600 genes. These genes were enriched for gene ontologies such as meiotic cell cycle, sexual reproduction, and sporulation, genes typically silenced in haploid yeast cells (Table S4). Indeed, the median transcripts per million (TPM) value of the down-regulated genes confirmed a low basal expression of the affected genes, with many of the silenced genes devoid of H3K4me3 or H3K36me3. Although H3K36me3 is typically associated with active transcription, it could be that at such heterochromatin loci it can also lead to further repression. Indeed, one of the roles of H3K36me3 within gene bodies is to prevent cryptic transcription and allow transcription initiation only at TSS, often marked by H3K4me3. The inhibition of cryptic transcription is achieved through recruitment of chromodomain protein Eaf3 that binds to trimethylated H3K36 and forms a part of an Rpd3-containing histone deacetylase complex, a known repressor of transcription initiation (Carrozza et al, 2005; Keogh et al, 2005). Interestingly, Rpd3 deacetylates histones at loci where it is recruited, leading to decreased spurious transcription, reduced RNA polymerase II recruitment and silencing of various genes, including meiotic transcripts (Kim et al, 2004; Lardenois et al, 2015). Therefore, an enhanced recruitment of Rpd3 complex components through increased H3K36me3 could be one possible explanation for the increased gene silencing observed in our data. Moreover, the increase in H3K36me3 at repressed loci could be associated with increased antisense transcription at the same loci. However, our preliminary analysis indicates that both sense and antisense transcripts are repressed at these loci (data not shown). Importantly, our results are consistent with the previous findings from Janke et al (2017) showing that inhibition of Rph1 or Gis1 by 2-HG can lead to increased silencing of a heterochromatin embedded reporter gene (Janke et al, 2017). Based on our data, this effect is more likely to depend on inhibition of Rph1. The presence of H3K36me3 in heterochromatin has also been reported for mutant IDH cells (Turcan et al, 2018) and recently has been identified to contribute to the regulation of heterochromatin in enhancers (Barral et al, 2022), suggesting a potential role for this modification in heterochromatin.

Taken together, our data point to a model where selected a-KG-dependent enzymes are preferentially inhibited by 2-HG resulting in changes in histone modifications that are largely decoupled from downstream effects on gene expression. Nevertheless, 2-HG-dependent gene expression changes can be observed at loci associated with a specific repressive chromatin state and could have implications on the role of 2-HG-mediated alterations in H3K36 methylation in human diseases.

# Materials and Methods

### Yeast strain and culturing

In this study, the wild-type *S. cerevisiae* strains used were of the BY4741 (*MATα*; *his3Δ1*; *leu2Δ0*; *met15Δ0*; *ura3Δ0*) or the FY4 background. The *DLD3* knockout strains were generated via PCR-mediated gene replacement. The KanMX cassette was used for the gene replacement as described previously (Becker-Kettern et al, 2016) and using, in this case, the following primer pairs for PCR amplification: forward ACATAATTGTCAAGAAAGCACAACA, reverse CGTGTTTAAAATTGTTGGATTAAGG. Regarding the histone demethylase deletion strains (*gis1Δ*, *rph1Δ*, *jhd1Δ*, *jhd2Δ*), all were in the BY4741 background and obtained from the collection of nonessential gene deletion mutants (complete yeast deletion strain set haploid MATa, Euroscarf). The full list of the strains used in this study can be found in the Table S5. *S. cerevisiae* was cultured in minimal media containing 1.7 g/liter YNB (114027512; MP biochemicals), 5 g/liter ammonium sulfate (A4418; Sigma-Aldrich), and 20 g/liter glucose (G8270; Sigma-Aldrich). For the BY4741 strains, the medium was supplemented with 80 mg/liter histidine (H8000; Sigma-Aldrich), methionine (M5308; Sigma-Aldrich) and uracil (U0750; Sigma-Aldrich) and 240 mg/liter leucine (L8912; Sigma-Aldrich). The pH was adjusted to 5.5 and the medium was sterilized by autoclaving or filtration. Yeast strains were cultured at 30°C and 200 rpm orbital shaking in shaking flasks and the working volume was 10% of the maximum allowed volume of the flask. The growth was measured by $OD_{600}$ monitoring in a microplate reader (Infinite M200Pro; Tecan) and the samples were collected when they reached the same point in the mid-exponential phase (0.6 optical density). The liquid cultures were inoculated from glycerol stocks of the yeast strains. The glycerol stocks had been prepared from single colonies cultured in minimal medium until stationary phase, followed by 10-fold concentration and addition of 20% glycerol. The stocks were stored at –80°C.

### ChIP

The *S. cerevisiae* strains were cultured until they reached the mid-exponential phase. Three biological replicates were generated for each strain and condition. The DNA histone crosslinking was achieved by incubation for 10 min with formaldehyde at a final concentration of 1%. The reaction was stopped with the addition of glycine (166 mM final concentration) and incubation for another 10 min. The cells were washed once with PBS and twice with PBS containing proteinase inhibitors (Complete Protease Inhibitor Cocktail, 11697498001; Merk). For the lysis of the cells, SDS lysis buffer (1% SDS, 10 mM EDTA, 80 mM Tris–HCl pH = 8) was used with the addition of proteinase inhibitors. The cells were first incubated for 12 min at room temperature followed by the addition of-0.5 mm glass beads (18406; Sigma-Aldrich) and processing with the Precellys 24 homogenizer (Bertin Technologies) for three cycles of 30 s on and 30 s off at max speed and 10–16°C. The DNA was fragmented using the Bioruptor Pico sonicator (Diagenode) set to sonicate at 8°C for 45 cycles, 30 s followed by 30 s pause. The fragmentation was checked by electrophoresis in a 1% agarose gel and only samples with fragments between 150–300 bp were selected. In this study, histone modifications were targeted using the anti-trimethyl-histone H3 lysine 4 (17-614; Merk-Millipore), the anti-trimethyl-histone H3 lysine 36 (ab9050; Abcam), and the anti-dimethyl-histone H3 lysine 36 (ab9049; Abcam) antibodies. For the anti-trimethyl-histone H3 lysine 4 immunoprecipitation reaction, 15 μg of chromatin were exposed to 3 μl of the antibody, according to the manufacturer's instructions. For the H3 lysine 36

immunoprecipitations, 25 µg of chromatin were exposed to 5 µg of each antibody. The samples were incubated with the antibodies for 6 h. For all immunoprecipitation reactions, 10% of the chromatin was used as input. After selecting the modifications with antibodies, 25 µl of PureProteome Protein A Magnetic Bead System (LSKMAGA10; Merk-Millipore) were used to isolate the chromatin–antibody complexes. The latter were washed with custom buffers, two times with wash buffer 1 (Tris–HCl 20 mM pH 8.0, NaCl 50 mM, EDTA 2 mM, TX-100 1%, SDS 0.1%), one time with wash buffer 2 (Tris–HCl 10 mM pH 8.0, NaCl 150 mM, EDTA 1 mM, NP-40 1%, sodium deoxycholate 1%, LiCl 250 mM), and two times with TE buffer (Tris–HCl 10 mM, EDTA 1 mM). Finally, the DNA was extracted at room temperature with the use of elution buffer (NaHCO3 0.1 M with SDS 1%), 20 µg recombinant proteinase K (AM2546; Thermo Fisher Scientific), and 10 µg RNase A (EN0531; Thermo Fisher Scientific). The DNA was purified by using the MinElute Reaction Cleanup Kit (28206; QIAGEN) according to the manufacturer's instructions and the DNA concentration was measured using the Qubit dsDNA HS Assay Kit (Q32851; Thermo Fisher Scientific).

## RNA extraction

All *S. cerevisiae* strains were cultured until the mid-exponential phase. For the FY4 and BY4741 strains, five and three biological replicates were used, respectively. 2 ml of cell culture was lysed with the addition of a buffer containing 1 M Sorbitol, 100 mM EDTA, and zymolyase (120493-1; Amsbio) at a final concentration of 100 U/ml and incubated for 30 min at 30°C. The RNeasy Mini Kit (74106; QIAGEN) was used to extract the RNA from the lysed cells according to the manufacturer's instructions for yeast cells. The RNA was further cleaned with the DNase Max Kit (15200-50; QIAGEN). The quality of the RNA was checked with the RNA 6000 Nano Kit (5067-1511; Agilent) and the 2100 Bioanalyzer. The RNA concentration was measured via the $A_{260}$ using the NanoDrop (ND-2000C; Thermo Fisher Scientific).

## Sequencing

All the sequencing data used in this study were generated at the sequencing platform of the Luxembourg Centre for Systems Biomedicine (LCSB), University of Luxembourg. For the RNA samples, libraries were generated by using the TruSeq Stranded mRNA Library Prep kit (20020594; Illumina) according to the manufacturer's instructions. The sequencing was performed by the Illumina NextSeq 500 machine with a read length of 75 bp. For the DNA samples collected by ChIP, single-end, reverse strand sequencing was applied by the Illumina NextSeq 500 machine with a read length of 75 bp. The sequencing data are available at ArrayExpress with accession number E-MTAB-11328.

## Metabolite extraction and liquid chromatography–mass spectrometry analysis

For metabolite extraction, all *S. cerevisiae* strains were cultured to the mid-exponential phase. Metabolite extraction and measurements were performed as previously described in a study by

Becker-Kettern et al (2016). Briefly, the 500 µl of cell culture were quenched by mixing the cell suspension with 60% cold methanol to generate pellets by centrifugation. For the extraction of metabolites, the pellets were mixed with 100-fold volume, measured with Multisizer 3 Coulter Counter (Beckman Counter), of cold extraction fluid (50% methanol, 50% TE buffer at pH 7.0 containing 10 mM Tris–HCl and 1 mM EDTA) and an equal volume of cold chloroform. After proper mixing, the upper aqueous phase was collected and stored at −20 EC. The concentration of total 2-HG was measured by the Luxembourg Centre for Systems Biomedicine metabolomics platform as described in the study by Becker-Kettern et al (2016).

## Bioinformatics analysis

### RNA-seq data analysis

The data generated by sequencing machines were checked for their quality with FastQC (https://www.bioinformatics.babraham.ac.uk/projects/fastqc/). The PALEOMIX pipeline (Schubert et al, 2014) (v 1.2.13.2) was used to trim reads using AdapterRemoval option (Schubert et al, 2016) with a minimal length of 35 bp (v2.2.3). The sequencing data were mapped to the S288C reference genome, assembly R64-2-1_20150113, downloaded from the Saccharomyces Genome Database (www.yeastgenome.org) using the splice-aware STAR aligner (Dobin et al, 2013) (v2.7.4a) with the following parameters:

--twopassMode Basic--outSAMtype BAM SortedByCoordinate.
--limitOutSJcollapsed 1000000 --limitSjdbInsertNsj 1000000.
--outFilterMultimapNmax 100 --outFilterMismatchNmax 33.
--outFilterMismatchNoverLmax 0.3 --seedSearchStartLmax 12 --alignSJoverhangMin 15.
--alignEndsType Local--outFilterMatchNminOverLread 0 --outFilterScoreMinOverLread 0.3.
--winAnchorMultimapNmax 50 --alignSJDBoverhangMin 3.

Read counting in the Yeast gene features was performed using featureCounts from RSubread (v2.8.1) (Liao et al, 2019) with default parameters, but with a minimal mapping quality of 30. Differential expression analysis was performed using the DESeq2 package (v1.32.0) (Love et al, 2014). As recommended by the authors, genes that had less than 10 counts across all samples were discarded. Log2FC shrinkage was performed using apeglm to preserve large effects of true positives (v1.14.0) (Zhu et al, 2019). TPM were obtained by using the fpkm() function in DESeq2 (after adding the gene lengths to the summarized experiment object). The resulting fpkm_yeast matrix was then converted to TPM using the following snippet: t(t[fpkm_yeast] * 1 × $10^6$/colSums[fpkm_yeast]) in R.

### ChIP-seq data analysis

The sequencing data generated from the ChIP samples were checked for their quality with FastQC (https://www.bioinformatics.babraham.ac.uk/projects/fastqc/). The PALEOMIX pipeline (Schubert et al, 2014) (v 1.2.13.2) was used to generate BAM files from the FASTQ files, including steps of adapter removal, mapping, and duplicate marking. The mapping was performed with the BWA option and the S288C reference genome was used similarly as for the RNA-seq data analysis. The H3K4me3 ChIP-seq peaks were called from the BAM files filtered for a minimal mapping quality of 30 with MACS2 (Zhang et al, 2008) (v 2.2.7.1) using the Narrow peak option, whereas for the H3K36me2 and H3K36me3 peaks, the Broad peak option was selected.

Finally, the mitochondrial DNA was removed. The peak reads were counted and normalized using Diffbind (Ross-Innes et al, 2012) (v 3.13). To count the reads, the function dba.count was used with the option summits set to false. For normalization, the reads were normalized for the library sizes by using the option DBA_NORM_LIB ("lib"). The final dba object was generated with the functions dba.contrast and dba.analyze without using any grey-listing option.

### Downstream analyses

**Metagene plots** The BAMs from the ChIP-seq, filtered for a minimal mapping quality of 30, were processed by bamCoverage (deepTools v3.5.0 [Ramírez et al, 2018]) using the CPM normalization to generate bigwig. Those bigwigs were further used as input for ComputeMatrix deepTools v3.5.0) using the following command scale-regions -m 1070 -b 150 -a 150 -bs 10 -R sc_R64-2-1_20150113.bed--metagene. This means that the CPM signal is binned per 10 bp, each gene's exons are merged, scaled to 1,070 bp (the median gene size across the yeast genome), and the flanking regions of 150 bp are also processed. The gene features were downloaded here: https://downloads.yeastgenome.org/ sequence/S288C_reference/genome_releases/S288C_reference_ genome_Current_Release.tgz similarly as for the STAR alignment (building the genome index step).

**PCA plots** Once the DESeq2 object was constructed from the read counts, the expression values were regularised using rlog() blind to the study design and then provided as input for the function plotPCA(). By default, the variance across all samples was computed and the top 500 genes were retained for the PCA.

**Correlation plots** For the ChIP-seq data, the function dba.report from Diffbind package (Ross-Innes et al, 2012) (v 3.13) was used to retrieve the FDR and the fold change for each peak. The package Tidygenomics (v 0.1.2) was used to identify the overlapping intervals between the peaks and the genes in the S288C reference genome by using the option genome_intersect. For the RNA-seq data, the FDR and the fold change were calculated with DESeq2. The plots were generated with ggplot2 (https://ggplot2.tidyverse.org/). The Pearson correlation was calculated with the function stat_cor from ggpubr package (v 0.4.0). For each of the correlation plots, the knockout genes of the compared strains (two at a time) were removed.

**Volcano plot** For the visualization of the differential binding sites, volcano plots were generated with the dba.plotVolcano function from Diffbind package (v 3.13). The sites with FDR lower than 0.05 and $\log_2$-fold change higher than 0.3 are highlighted with red colour.

**IGV images** The BED files generated from MACS2 analysis were converted to bigwig files with bedGraphToBigWig (Kent et al, 2010). The generated files were loaded to the IGV (Robinson et al, 2011) (v 2.9.1).

### Expression and purification of recombinant enzymes

The coding sequence of the full-length yeast demethylases Rph1 (GenBank: KZV11945.1), Gis1 (GenBank: KZV12332.1), Jhd1 (GenBank: QHB08122.1), and Jhd2 (GenBank: QHB09742.1) were cloned into the pET51b+ plasmid for expression of recombinant fusion proteins containing an N-terminal Strep(II) affinity tag as suggested by Krishnan et al (2012). The *Pseudomonas putida* formaldehyde dehydrogenase (FDH) (UniProt ID P46154.3) was cloned into the pet28a+ plasmid for expression of C-terminally His-tagged protein as described by Roy & Bhagwat (2007). All the sequences were codon-optimized for *Escherichia coli* expression by the Genscript codon optimization tool.

Recombinant plasmids were transformed into One Shot BL21 Star (DE3) chemically competent *E. coli* cells (601003; Thermo Fisher Scientific) according to the manufacturer's protocol and plated on solid LB medium containing ampicillin (50 $\mu$g/ml). After 24 h, single colonies were inoculated for overnight culture in 5 ml LB medium containing ampicillin (50 $\mu$g/ml) and plasmids were purified (QIAprep Spin Miniprep Kit, 27106) and sent to sequencing to confirm the presence of our gene of interest.

For recombinant protein production, single colonies were inoculated in 50 ml LB medium containing ampicillin (50 $\mu$g/ml) and incubated overnight at 37°C with shaking (220 rpm). The culture was diluted to an $OD_{600}$ of 0.1 in 3 liters of the same medium and cultivated under the same conditions until $OD_{600}$ reached 0.8. The culture was cooled on ice for 30 min, followed by the addition of IPTG to 1.0 mM final concentration. Cells were further cultivated at 18°C with shaking at 80 rpm for 16–18 h and pelleted by centrifugation (16,000 rpm and 4°C). The pellets were resuspended in 50 ml of freshly made lysis buffer (25 mM Tris pH 8, 300 mM NaCl, 0.5 mM PMSF, 1 mM DTT, and 1 "cOmplete ULTRA Tablet, Mini, EDTA-free, EASYpack Protease Inhibitor Cocktail inhibitors" from Roche, 04693159001), sonicated on ice, and treated with DNAse I (100 $\mu$g/ml final concentration) and $MgSO_4$ (1 mM final concentration). The solution was pre-clarified by centrifugation for 30 min at 10,000 rpm and 4°C, and the supernatant was filtered on a 0.45-$\mu$m syringe filter.

### Protein purification

Proteins were purified on an Akta purification system (Unicorn 7.5 software) using 1 ml StrepTrap HP columns (28907547; Cytiva) for the demethylases, and a 5 ml HisTrap HP column (524802; Cytiva) for FDH. Briefly, columns were washed with 20 column volumes (CV) degassed Milli-Q water at a flow rate of 1 ml/min and equilibrated with 20 CVs of 0.45-$\mu$m filtered and degassed buffer A (20 mM $Na_3PO_4$, 280 mM NaCl, 6 mM KCl, pH 7.4 for StrepTrap HP columns, and 25 mM Tris, 300 mM NaCl, 10 mM imidazole, pH 8.0 for HisTrap HP columns), also at 1 ml/min. Clarified and filtered lysates from the overexpression cells were supplemented with buffer A components to reach the same final concentrations as in buffer A. The diluted lysates were loaded onto the equilibrated column at 0.5 ml/min, followed by a washing step with Buffer A (plus 10 mM imidazole in the case of FDH purification) at 1 ml/min to remove nonspecifically bound proteins, until the $A_{280}$ was back to a zero value.

The His-tagged FDH protein was eluted by applying a gradient from 3 to 100% buffer B (25 mM Tris, 300 mM NaCl, 300 mM Imidazole, pH 8.0) over 20 ml. Protein peak fractions were collected and separately analysed by Western blotting to confirm the presence of our protein of interest. HiTrap desalting columns (17140801; Cytiva) were used to remove excess imidazole and Vivaspin protein

**Life Science Alliance**

concentrator spin columns (VS0102; Sartorius) were used to concentrate the pooled fractions of interest.

Strep-tagged proteins were eluted by applying 20 CVs of 100% filtered and degassed buffer C (20 mM Na$_3$PO$_4$, 280 mM NaCl, 6 mM KCl, 2.5 mM desthiobiotin, pH 7.4) at 1 ml/min, with the protein of interest eluting in a single peak between 8 and 12 CVs. Peak fractions were analysed through Western blotting to confirm the protein identity. Vivaspin protein concentrator spin columns (VS0102; Sartorius) were used to increase protein concentration.

### FDH-coupled fluorescent histone demethylase activity assay

An FDH-coupled fluorescent assay was used to measure histone demethylase activity. The histone demethylase reaction releases formaldehyde which is then oxidized by FDH to formic acid with concomitant conversion of NAD+ to NADH, whose fluorescence was measured in a Tecan microplate reader (excitation and emission wavelengths set to 340 nm and 480 nm, respectively). The FDH-coupled fluorescent assay was performed as described by Krishnan & Trievel (2016) with some modifications. The final concentrations of FDH and methylated peptides (Anaspec, AS-64380-1, AS-64381-1, AS-64636-1, AS-64379-1) was 1 $\mu$M and 1 mM, respectively, and a range of 2-HG concentrations was tested. Buffer 1 (final concentrations of 50 mM HEPES pH 7.5, 50 mM NaCl, 50 $\mu$M ammonium iron (II) sulfate, 1 mM L-ascorbic acid, 1 mM NAD+, 1 $\mu$M FDH, 1.5 $\mu$M demethylase of interest) was loaded into a 96-well plate already placed in a warmed-up microplate reader and the reaction was started by adding Buffer 2 (final concentrations of 1 mM a-KG, 0–10 mM 2-HG, 1 mM methylated peptide substrate).

The relative fluorescence unit output from the Tecan microplate reader was converted into NADH concentrations based on a calibration curve with a standard NADH solution. Control reactions in which FDH was omitted were run in each experiment and used for background corrections of the fluorescence values obtained with the complete reaction mixtures. Initial velocities (in $\mu$M NADH produced per min) were determined over a period of 1.5 min and the final activities presented represent means of three independent replicates.

### IC50 estimation

A Hill Equation sigmoid was fitted to the vectors of concentrations and respective responses per enzyme and peptides using a slightly modified version of the doseResponse function in MATLAB (https://www.mathworks.com/matlabcentral/fileexchange/33604-doseresponse, retrieved from MATLAB Central File Exchange on November 9, 2023). The 2-HG concentration where the curves were yielding 50% of the difference between the fitted values of the minimal and maximal experimental concentrations was estimated and reported as IC50 value per enzyme and peptide.

## Data Availability

Raw FASTQ files were deposited in ArrayExpress with accession number E-MTAB-11328.

## Supplementary Information

## Acknowledgements

We would like to thank Dean Cheung for his assistance in metabolite extraction. The computational analyses presented in this article were carried out using the HPC facilities of the University of Luxembourg. L Sinkkonen and CL Linster would like to thank the Internal Research Project (IRP) funding from the University of Luxembourg for financial support. Nicole Paczia was supported by FNR Core Junior funding (C16/BM/11339953).

### Author Contributions

M Gavriil: conceptualization, data curation, formal analysis, investigation, visualization, methodology, and writing—original draft, review, and editing.
M Proietto: formal analysis, investigation, visualization, methodology, and writing—review and editing.
N Paczia: conceptualization, formal analysis, supervision, funding acquisition, investigation, and writing—review and editing.
A Ginolhac: data curation, software, formal analysis, visualization, methodology, and writing—review and editing.
R Halder: formal analysis, methodology, and writing—review and editing.
E Valceschini: formal analysis, investigation, methodology, and writing—review and editing.
T Sauter: formal analysis, supervision, funding acquisition, investigation, project administration, and writing—review and editing.
CL Linster: conceptualization, resources, formal analysis, supervision, funding acquisition, project administration, and writing—original draft, review, and editing.
L Sinkkonen: conceptualization, supervision, funding acquisition, investigation, project administration, and writing—original draft, review, and editing.

### Conflict of Interest Statement

The authors declare that they have no conflict of interest.

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
