## [Reviewer comments · Life Science Alliance]

Life Science Alliance

2-HG modulates histone methylation at specific loci and alters gene expression via Rph1 inhibition

Marios Gavriil, Marco Proietto, Nicole Paczia, Aurélien Ginolhac, Rashi Halder, Elena Valceschini, Thomas Sauter, Carole Linster, and Lasse Sinkkonen

DOI: <https://doi.org/10.26508/lsa.202302333>

Corresponding author(s): Lasse Sinkkonen, University of Luxembourg and Carole Linster, University of Luxembourg

Review Timeline:

Submission Date:	2023-08-23
Editorial Decision:	2023-09-19
Revision Received:	2023-11-15
Editorial Decision:	2023-11-20
Revision Received:	2023-11-21
Accepted:	2023-11-21

Transaction Report:

September 19, 2023

Re: Life Science Alliance manuscript #LSA-2023-02333-T

Dr. Lasse Sinkkonen
University of Luxembourg
Department of Life Science and Medicine
6, Avenue du Swing
Belvaux L-4367
Luxembourg

Dear Dr. Sinkkonen,

Thank you for submitting your manuscript entitled "2-HG modulates histone methylation at specific loci and alters gene expression via Rph1 inhibition" to Life Science Alliance. The manuscript was assessed by expert reviewers, whose comments are appended to this letter. We invite you to submit a revised manuscript addressing the Reviewer comments.

Thank you for this interesting contribution to Life Science Alliance. We are looking forward to receiving your revised manuscript.

Sincerely,

B. MANUSCRIPT ORGANIZATION AND FORMATTING:

Reviewer #1 (Comments to the Authors (Required)):

Dr. Sinkkonen and coauthors have investigated the consequences of 2-HG accumulation in *S. cerevisiae* concentrating of epigenetic analyses. The work is well-planned and thorough and brings new knowledge to the field. They identified Rph1, the yeast homolog of human KDM4s, being the most sensitive to the inhibitory effect of 2-HG in vitro. Quite unexpectedly, Rph1 deficiency associated with gene repression and led to further downregulation of already silenced genes marked by low H3K4 and H3K36 trimethylation but high H3K36 dimethylation. In my opinion the work deserves to be published but I have a few issues I would like the authors to respond.

1. The methylation status of a loci is a balance between the activities of methyl transferases and demethylases. How are the results presented controlled for methyl transferase activity?
2. Have the authors considered could they present the data on inhibition of the yeast methyltransferases by 2-HG in IC50 values which would be more comparable and highly valuable for the larger scientific community?

Minor

1. Correct the format of Reference #13

Reviewer #2 (Comments to the Authors (Required)):

This manuscript investigates the effects of increasing 2-HG levels on histone modification and gene expression in *Saccharomyces cerevisiae*. The selected model system has the advantage to be devoid of DNA methylation and repressive histone methylation, so that the investigation can be focused on H3K4 and H3K36 methylation levels and eventual gene expression changes associated with them. The Authors also present a genome-wide analysis of H3K4 and H3K36 methylation on strains deleted on the individual histone demethylases which shows very interesting results.

The main conclusion is that Rph1 is the yeast demethylase most sensitive to the inhibitory effect of 2-HG in vivo (well supported by data), probably reflecting its prominent in vitro sensitivity (this part needs to be further detailed, see main points below). Interestingly, Rph1 deficiency favors gene repression and leads to further downregulation of already silenced genes marked by low H3K4 and H3K36 trimethylation, but abundant in H3K36 dimethylation. The work is sound, the experiments are well designed and complete and clearly presented. The results may give a relevant contribution to a better understanding of the effects of altered 2-HG levels on the epigenetic regulation of eukaryotic cells. For these reasons, the manuscript deserves publication on Life Science Alliance.

On the other hand, there are a couple of main points which should be addressed prior to publication in order to improve the clarity of the text and strength the conclusions.

Main points

-The Authors observe an increase of around 20-fold of the 2-HG intracellular concentration in *dld3* deleted strains in both genetic backgrounds as compared with the corresponding wt strains. On the other hand, they don't mention the intranuclear concentration which is probably more relevant. Also the α -KG intranuclear concentration (which should be also lowered by the *dld3* deletion) is a relevant parameter. If the Authors want to support their conclusion that Rph1 is the histone demethylase most affected by 2-HG in vivo due to its higher sensitivity to the inhibitor, they should show that the in vitro assay is performed at α -KG concentrations similar to the nuclear concentration in vivo (the different enzymes have different KM for α -KG).

-The most puzzling results of the work are represented by ChIPseq results. In the *delta-jhd2* strain the number of loci which show decreased H3K4 tri-methylation is twice the number of those which show increased tri-methylation. In the case of *delta-rph1* H3K36 tri-methylation significantly decreases in 710 loci and increases in 104 only. This paradoxical results need to be addressed and discussed. Possible explanations are:

- Feedback regulation (repression) of cognate histone methylases abundance in the absence of demethylases
- Regulation of cognate histone methylases activity by co-localization with histone demethylases
- Regulation of histone methylases targeting by availability of demethylated substrate

This hypotheses were previously proposed and discussed in several papers including:

1. Ramakrishnan, S.; Pokhrel, S.; Palani, S.; Pflueger, C.; Parnell, T.J.; Cairns, B.R.; Bhaskara, S.; Chandrasekharan, M.B. Counteracting H3K4 Methylation Modulators Set1 and Jhd2 Co-Regulate Chromatin Dynamics and Gene Transcription. *Nat.*

Commun. 2016, 7, 11949.

2. Soares, L.M.; Radman-Livaja, M.; Lin, S.G.; Rando, O.J.; Buratowski, S. Feedback Control of Set1 Protein Levels Is Important for Proper H3K4 Methylation Patterns. *Cell Rep.* 2014, 6, 961-972.

3. Di Nisio, E.; Danovska, S.; Condemni, L.; Cirigliano, A.; Rinaldi, T.; Licursi, V.; Negri, R. H3 Lysine 4 Methylation Is Required for Full Activation of Genes Involved in α -Ketoglutarate Availability in the Nucleus of Yeast Cells after Diauxic Shift. *Metabolites* 2023, 13, 507. <https://doi.org/10.3390/metabo13040507>

The complexity of histone methylase/demethylase dynamics should be taken in account when discussing the effects of 2-HG induced inhibition on gene expression.

Minor points:

There are several typos (especially in the references).

'Referee Cross-Comments' to your review report.

Reviewer 1's comment 1: "The methylation status of a loci is a balance between the activities of methyl transferases and demethylases. How are the results presented controlled for methyl transferase activity?" is on line with my request to discuss the possible feedback effects of histone demethylase deletion/inhibition on methylase activity. This does not need to be done experimentally but it is a crucial point to be addressed.

Authors' Response to Reviewers

Manuscript number: LSA-2023-02333-T

Corresponding author(s): Sinkkonen, Lasse and Linster, Carole

General Statements

We would like to thank you for considering our manuscript for publication in Life Science Alliance and appreciate the very useful feedback from the reviewers.

Point-by-point response to reviewers

Reviewer #1

Dr. Sinkkonen and coauthors have investigated the consequences of 2-HG accumulation in *S. cerevisiae* concentrating of epigenetic analyses. The work is well-planned and thorough and brings new knowledge to the field. They identified Rph1, the yeast homolog of human KDM4s, being the most sensitive to the inhibitory effect of 2-HG in vitro. Quite unexpectedly, Rph1 deficiency associated with gene repression and led to further downregulation of already silenced genes marked by low H3K4 and H3K36 trimethylation but high H3K36 dimethylation. In my opinion the work deserves to be published but I have a few issues I would like the authors to respond.

Thank you to Reviewer 1 for the very useful suggestions that have helped us to improve the manuscript as detailed below. We have addressed these comments through changes in the manuscript text in the Results and Discussion sections, by estimating IC50 values for 2-HG inhibition of the tested enzymes, and addition of new panels B and C in Supplementary Figure 2 as detailed below. All new text in the manuscript is in red font.

1. The methylation status of a loci is a balance between the activities of methyl transferases and demethylases. How are the results presented controlled for methyl transferase activity?

Thank you for highlighting the potential role of methyltransferases in our findings and its possible implications on the seemingly contradicting results regarding hyper- and hypomethylation in our mutant strains. In our histone demethylase knockouts, we do observe an overall accumulation of methylation in their target modifications. To allow more complete representation of our data, we have added new panels B and C in the Supplementary Figure 2. These metagenepLOTS show that we can observe an increased median H3K4me3 signal in the BY4741 *jhd2Δ* strain, compared to the wild-type, when focusing on the signal across all genes (panel B, please find the new panels below). Similarly the H3K36me3 signal is increased in the BY4741 *rph1Δ* strain (panel C). Indicating that an overall increase in methylation is observed in both cases as could be expected upon demethylase deletion.

For more clarity, text changes have been made in the Results section on pages 6-9 of the revised manuscript. These results align with other publications where they investigated bulk histone methylation changes upon deletion of histone demethylases in yeast (Tu S *et al.*, J Biol Chem 2007, doi:10.1074/jbc.M609900200).

On the other hand, following a differential peak analysis using MACS2 and DiffBind, the number of hypomethylated DMGs in BY4741 *dld3Δ* cells was higher than the hypermethylated DMGs for both H3K36 modifications. Similarly, there was a greater number of hypomethylated DMGs, for both H3K4me3 and H3K36me3, compared to the number of hypermethylated ones in both *jhd2Δ* and *rph1Δ* cells. To address this apparent contradiction, to discuss the possible explanations, and to highlight the putative role of methyltransferases, we have added a new paragraph in the Discussion section on page 13 of the revised manuscript. For your convenience, please find the new paragraph here:

*In this study we have utilized genetic knockout strains that have adapted to the loss of the deleted enzymes. Since histone methylation is a balance of the activities of methyltransferases and demethylases, our results do not necessarily reflect the primary changes occurring upon inhibition or deletion of demethylases but rather the acquired steady states. Observing the median methylation signals across all genes for 2-HG accumulating *dld3Δ* cells revealed an expected change of increased H3K4me3 and H3K36me3 at the expense of reduced H3K36me2 (Figure 2A) Similarly, deletion of specific histone demethylases led to an anticipated accumulation of methylation across genes (Supplementary Figure 2). However, a statistical identification of DMGs with DiffBind revealed a different picture with variable responses at individual loci (Figure 2B). This apparent contradiction can be partially explained by the procedure of DMG identification that requires the presence of a sufficient signal to be called as a peak by MACS2, biasing the analysis to loci with already detectable methylation,*

while an increase in methylation was observed in particular at downregulated genes with low levels of H3K4 or H3K36 methylation (see Methods for details). In addition to this technical explanation, also other biological mechanisms could explain why more of the already methylated loci showed reduced, rather than increased methylation upon the deletion of the corresponding demethylase (Figure 3-5).

One possible explanation could be the simultaneous downregulation of histone methyltransferases in response to histone demethylase deletion. However, none of the relevant methyltransferases (*Set2*, *Set2*, and *Dot1*) or the members of the COMPASS complex changed in their mRNA expression in any of the histone demethylase deletion strains (Supplementary Tables S1 and S4). Nevertheless, we cannot exclude changes in protein abundance or localization that could affect the compensation of loss of demethylation differently at different loci (Soares et al, 2014).

Other possibilities include an altered availability of the methyl-donor or interdependence between methylation and demethylation processes. It has been previously shown that a feedback control exists between expression of enzymes controlling α -KG availability and demethylase activity (Di Nisio et al, 2023). A similar feedback could also exist for availability of S-adenosylmethionine (SAM) for methylation, although our transcriptome data indicated no changes in the expression of *Sam1* or *Sam2*, enzymes responsible for SAM synthesis in yeast (Supplementary Tables S1 and S4). Still, a feedback between demethylase and methyltransferase activities, and co-regulation of their target genes, have been reported before for H3K4me3 (Ramakrishnan et al, 2016; Di Nisio et al, 2023). In detail, a loss of either the methyltransferase or the demethylase can lead to comparable changes in target gene expression, depending on the chromatin context, although opposing outcomes would be expected. Such explanation might exist for differential responses to loss of demethylase activity also for other targets like H3K36me3. Lastly, it has been shown that yeast cells can regulate histone methylation levels and respond to stimuli, such as carbon source availability, through DNA replication and cell cycle (Sein Henel AND Värnv, 2015). It could be that our yeast strains have adapted their methylation levels through several cell cycles.

2. Have the authors considered could they present the data on inhibition of the yeast methyltransferases by 2-HG in IC50 values which would be more comparable and highly valuable for the larger scientific community?

Thank you for this interesting suggestion. We have now estimated the IC50 values for 2HG inhibition of all the studied demethylases per peptide and added them to the manuscript text in the Results section of the revised manuscript. Details for the IC50 estimation based on our *in vitro* enzyme activity assays are provided in the Methods section on page 24 of revised manuscript. The obtained IC50 values were 3.1-5.0 mM (*Gis1*), 2.0 mM (*Jhd1*), 2.2 mM (*Jhd2*), and 1.3-1.5 mM (*Rph1*).

Correct the format of Reference #13

Thank you for careful reading of the manuscript and identifying the error. The format of the reference has been corrected in the revised version of the manuscript.

Reviewer #2

This manuscript investigates the effects of increasing 2-HG levels on histone modification and gene expression in *Saccharomyces cerevisiae*. The selected model system has the advantage to be devoid of DNA methylation and repressive histone methylation, so that the investigation can be focused on H3K4 and H3K36 methylation levels and eventual gene expression changes associated with them. The Authors also present a genome-wide analysis of H3K4 and H3K36 methylation on strains deleted on the individual histone demethylases which shows very interesting results.

The main conclusion is that Rph1 is the yeast demethylase most sensitive to the inhibitory effect of 2-HG in vivo (well supported by data), probably reflecting its prominent in vitro sensitivity (this part needs to be further detailed, see main points below). Interestingly, Rph1 deficiency favors gene repression and leads to further downregulation of already silenced genes marked by low H3K4 and H3K36 trimethylation, but abundant in H3K36 dimethylation. The work is sound, the experiments are well designed and complete and clearly presented. The results may give a relevant contribution to a better understanding of the effects of altered 2-HG levels on the epigenetic regulation of eukaryotic cells. For these reasons, the manuscript deserves publication on Life Science Alliance.

On the other hand, there are a couple of main points which should be addressed prior to publication in order to improve the clarity of the text and strength the conclusions.

Thank you to Reviewer 2 for highlighting that some of the conclusions in our study require clarification. We have addressed these comments through changes in the manuscript text in the Results and Discussion sections and addition of new panels B and C in Supplementary Figure 2 as detailed below. In addition, we have estimated IC50 values for 2-HG inhibition of the tested enzymes as requested by Reviewer 1. All new text in the manuscript is in red font.

-The Authors observe an increase of around 20-fold of the 2-HG intracellular concentration in *dld3* deleted strains in both genetic backgrounds as compared with the corresponding wt strains. On the other hand, they don't mention the intranuclear concentration which is probably more relevant. Also the α -KG intranuclear concentration (which should be also lowered by the *dld3* deletion) is a relevant parameter. If the Authors want to support their conclusion that Rph1 is the histone demethylase most affected by 2-HG in vivo due to its higher sensitivity to the inhibitor, they should show that the in vitro assay is performed at α -KG concentrations similar to the nuclear concentration in vivo (the different enzymes have different K_M for α -KG).

We thank the Reviewer for this valuable comment. We could not find K_M values for Rph1 or the other yeast demethylases studied here in the literature or in the Brenda database, and we did not determine these values ourselves. An intracellular α -KG concentration of 0.44 mM was previously reported for *E. coli* cells by Bennett *et al.* (Nature Chem Biol 2009, doi: 10.1038/nchembio.186). For *S. cerevisiae* cells, we found intracellular α -KG concentrations of around 0.1 mM and *dld3* deletion did not change these intracellular concentrations significantly (only extracellular α -KG levels were decreased in cultures of cells accumulating 2-HG compared to cultures of wild-type cells), despite the 20-fold increase in intracellular 2-HG concentrations in the *dld3* KO strain (Becker-Kettern *et al.*, J Biol Chem 2016; Fig. 2 and

supplem. Figs. S2 and S3; doi:10.1074/jbc.M115.704494). Metabolites can diffuse freely across the nuclear pores and it is generally assumed that intranuclear concentrations are similar to cytoplasmic concentrations (Cambronne *et al.*, Science 2016; doi:10.1126/science.aad5168). In our demethylase activity assays we added a-KG and the methylated peptide substrate at a final concentration of 1 mM. This choice was based on previous work where assay parameters were optimized (including for a-KG concentration) for the human JMJD2 lysine demethylases (Khrisnan *et al.*, Anal Biochem, doi:10.1016/j.ab.2011.08.034). Assuming that the K_m of the various demethylases for a-KG fluctuates around the intracellular a-KG concentration, the recombinant demethylases should be saturated by this co-substrate in our *in vitro* assays. This has the advantage of yielding more reproducible activity measurements, given generally greater experimental errors in kinetic measurements when working at subsaturating cofactor concentrations.

In conclusion, while we agree with the Reviewer that additional and perhaps physiologically more relevant insights could be gained when performing the *in vitro* 2-HG inhibition studies also at lower concentrations of aKG, we provide here a first comparative analysis of this 2-HG effect on the demethylase isoforms at saturating cosubstrate concentration, which is an experimental condition allowing for more robust measurements. Under these conditions, our results indicate that Rph1 is more sensitive to 2-HG inhibition ($IC_{50} = 1.3-1.5$ mM) than the other yeast demethylases ($IC_{50} > 2.0$ mM), which supports the *in vivo* results.

In the revised manuscript, we specified in the Results section (p. 8-9 of revised manuscript) that our conclusion on higher sensitivity of Rph1 to 2-HG inhibition *in vitro* is only valid under the assay conditions used. We also added a sentence to encourage future work to refine the *in vitro* enzyme characterization by repeating the inhibition studies at lower, near-physiological concentrations of a-KG. If the K_m of the demethylases is close to this physiological concentration, different (likely lower) IC_{50} 's for 2-HG may be obtained. If, however, the K_m is lower than the physiological concentration of a-KG (or under conditions of elevated intracellular a-KG concentrations), the demethylases would be saturated by this cosubstrate also in the cell and the IC_{50} 's reported in this manuscript based on our *in vitro* assays should approximate the intracellular ones.

-The most puzzling results of the work are represented by ChIPseq results. In the delta-jhd2 strain the number of loci which show decreased H3K4 tri-methylation is twice the number of those which show increased tri-methylation. In the case of delta-rph1 H3K36 tri-methylation significantly decreases in 710 loci and increases in 104 only. This paradoxical results need to be addressed and discussed. Possible explanations are:

- Feedback regulation (repression) of cognate histone methylases abundance in the absence of demethylases
- Regulation of cognate histone methylases activity by co-localization with histone demethylases
- Regulation of histone methylases targeting by availability of demethylated substrate

This hypotheses were previously proposed and discussed in several papers including:

1.Ramakrishnan, S.; Pokhrel, S.; Palani, S.; Pflueger, C.; Parnell, T.J.; Cairns, B.R.; Bhaskara, S.; Chandrasekharan, M.B. Counteracting H3K4 Methylation Modulators Set1 and Jhd2 Co-

Regulate Chromatin Dynamics and Gene Transcription. Nat. Commun. 2016, 7, 11949.
 2. Soares, L.M.; Radman-Livaja, M.; Lin, S.G.; Rando, O.J.; Buratowski, S. Feedback Control of Set1 Protein Levels Is Important for Proper H3K4 Methylation Patterns. Cell Rep. 2014, 6, 961-972.

3. Di Nisio, E.; Danovska, S.; Condemni, L.; Cirigliano, A.; Rinaldi, T.; Licursi, V.; Negri, R. H3 Lysine 4 Methylation Is Required for Full Activation of Genes Involved in α -Ketoglutarate Availability in the Nucleus of Yeast Cells after Diauxic Shift. Metabolites 2023, 13, 507. <https://doi.org/10.3390/metabo13040507>

The complexity of histone methylase/demethylase dynamics should be taken in account when discussing the effects of 2-HG induced inhibition on gene expression.

We would like to thank Reviewer 2 for highlighting a misleading presentation of our data that required changes in our text. In our histone demethylase knockouts, we can observe an overall accumulation of methylation in their target modifications. To better address this fact we have added new panels B and C in the Supplementary Figure 2. These metagenepLOTS show that we can observe an increased median H3K4me3 signal in the BY4741 *jhd2* Δ strain, compared to the wild-type, when focusing on the signal across all genes (panel B, please find the new panels below). Similarly the H3K36me3 signal is increased in the BY4741 *rph1* Δ strain (panel C). Indicating that an overall increase in methylation is observed in both cases as could be expected upon demethylase deletion.

B

C

For more clarity, text changes have been made in the Results section on pages 6-9 of the revised manuscript. These results align with other publications where they investigated bulk histone methylation changes upon deletion of histone demethylases in yeast (Tu S *et al.*, J Biol Chem 2007, doi:10.1074/jbc.M609900200).

On the other hand, following a differential peak analysis using MACS2 and DiffBind, the number of hypomethylated DMGs in BY4741 *dld3Δ* cells was higher than the hypermethylated DMGs for both H3K36 modifications. Similarly, there was a greater number of hypomethylated DMGs, for both H3K4me3 and H3K36me3, compared to the number of hypermethylated ones in both *jhd2Δ* and *rph1Δ* cells. To address this apparent contradiction and to discuss the possible explanations, we have added a new paragraph in the Discussion section on page 13 of the revised manuscript. For your convenience, please find the new paragraph here:

*In this study we have utilized genetic knockout strains that have adapted to the loss of the deleted enzymes. Since histone methylation is a balance of the activities of methyltransferases and demethylases, our results do not necessarily reflect the primary changes occurring upon inhibition or deletion of demethylases but rather the acquired steady states. Observing the median methylation signals across all genes for 2-HG accumulating *dld3Δ* cells revealed an expected change of increased H3K4me3 and H3K36me3 at the expense of reduced H3K36me2 (Figure 2A) Similarly, deletion of specific histone demethylases led to an anticipated accumulation of methylation across genes (Supplementary Figure 2). However, a statistical identification of DMGs with DiffBind revealed a different picture with variable responses at individual loci (Figure 2B). This apparent contradiction can be partially explained by the procedure of DMG identification that requires the presence of a sufficient signal to be called as a peak by MACS2, biasing the analysis to loci with already detectable methylation, while an increase in methylation was observed in particular at downregulated genes with low levels of H3K4 or H3K36 methylation (see Methods for details). In addition to this technical explanation, also other biological mechanisms could explain why more of the already methylated loci showed reduced, rather than increased methylation upon the deletion of the corresponding demethylase (Figure 3-5).*

*One possible explanation could be the simultaneous downregulation of histone methyltransferases in response to histone demethylase deletion. However, none of the relevant methyltransferases (*Set2*, *Set2*, and *Dot1*) or the members of the COMPASS complex changed in their mRNA expression in any of the histone demethylase deletion strains (Supplementary Tables S1 and S4). Nevertheless, we cannot exclude changes in protein abundance or localization that could affect the compensation of loss of demethylation differently at different loci (Soares et al, 2014).*

*Other possibilities include an altered availability of the methyl-donor or interdependence between methylation and demethylation processes. It has been previously shown that a feedback control exists between expression of enzymes controlling α -KG availability and demethylase activity (Di Nisio et al, 2023). A similar feedback could also exist for availability of S-adenosylmethionine (SAM) for methylation, although our transcriptome data indicated no changes in the expression of *Sam1* or *Sam2*, enzymes responsible for SAM synthesis in yeast (Supplementary Tables S1 and S4). Still, a feedback between demethylase and methyltransferase activities, and co-regulation of their target genes, have been reported before for H3K4me3 (Ramakrishnan et al, 2016; Di Nisio et al, 2023). In detail, a loss of either the methyltransferase or the demethylase can lead to comparable changes in target gene expression, depending on the chromatin context, although opposing outcomes would be expected. Such explanation might exist for differential responses to loss of demethylase activity also for other targets like H3K36me3. Lastly, it has been shown that yeast cells can regulate histone methylation levels and respond to stimuli, such as carbon source availability,*

through DNA replication and cell cycle (Sein Henel AND Värvi, 2015). It could be that our yeast strains have adapted their methylation levels through several cell cycles.

There are several typos (especially in the references).

Thank you for careful reading of the manuscript and identifying the typos. We have aimed to correct all typos in the revised version of the manuscript.

Reviewer 1's comment 1: "The methylation status of a loci is a balance between the activities of methyl transferases and demethylases. How are the results presented controlled for methyl transferase activity?" is on line with my request to discuss the possible feedback effects of histone demethylase deletion/inhibition on methylase activity. This does not need to be done experimentally but it is a crucial point to be addressed.

Thank you for your useful suggestion that we have followed in the revised manuscript.

November 20, 2023

RE: Life Science Alliance Manuscript #LSA-2023-02333-TR

Dr. Lasse Sinkkonen
University of Luxembourg
Department of Life Sciences and Medicine
6, Avenue du Swing
Belvaux L-4367
Luxembourg

Dear Dr. Sinkkonen,

Thank you for submitting your revised manuscript entitled "2-HG modulates histone methylation at specific loci and alters gene expression via Rph1 inhibition". We would be happy to publish your paper in Life Science Alliance pending final revisions necessary to meet our formatting guidelines.

- please upload all figure files as individual ones, including the supplementary figure files
- please add your main, supplementary figure, and table legends to the main manuscript text after the references section
- please add the Twitter handle of your host institute/organization as well as your own or/and one of the authors in our system
- please update your callouts for the Supplementary Figures in the manuscript text Fig S1 (a,b,c,d,e,f,g,h), Fig S2 (a,b,c)

A. FINAL FILES:

B. MANUSCRIPT ORGANIZATION AND FORMATTING:

Sincerely,

Reviewer #2 (Comments to the Authors (Required)):

The Authors have convincingly addressed each and every point which I raised in my review. I found the manuscript improved and ready to be accepted for publication on Life Science Alliance.

November 21, 2023

RE: Life Science Alliance Manuscript #LSA-2023-02333-TRR

Dr. Lasse Sinkkonen
University of Luxembourg
Department of Life Sciences and Medicine
6, Avenue du Swing
Belvaux L-4367
Luxembourg

Dear Dr. Sinkkonen,

Thank you for submitting your Research Article entitled "2-HG modulates histone methylation at specific loci and alters gene expression via Rph1 inhibition". It is a pleasure to let you know that your manuscript is now accepted for publication in Life Science Alliance. Congratulations on this interesting work.

DISTRIBUTION OF MATERIALS:

Again, congratulations on a very nice paper. I hope you found the review process to be constructive and are pleased with how the manuscript was handled editorially. We look forward to future exciting submissions from your lab.

Sincerely,
